# GPO: Learning from Critical Steps to Improve LLM Reasoning

**Jiahao Yu**
Department of Computer Science
Northwestern University
jiahao.yu@northwestern.edu

**Zelei Cheng**[*]
AI Foundations
Capital One
Department of Computer Science
Northwestern University
zelei.cheng@capitalone.com

**Xian Wu**[†]
Meta AI
xianwu123@meta.com

**Xinyu Xing**[†]
Department of Computer Science
Northwestern University
xinyu.xing@northwestern.edu

## Abstract

Large language models (LLMs) are increasingly used in various domains, showing impressive potential on different tasks. Recently, reasoning LLMs have been proposed to improve the *reasoning* or *thinking* capabilities of LLMs to solve complex problems. Despite the promising results of reasoning LLMs, enhancing the multi-step reasoning capabilities of LLMs still remains a significant challenge. While existing optimization methods have advanced the LLM reasoning capabilities, they often treat reasoning trajectories as a whole, without considering the underlying critical steps within the trajectory. In this paper, we introduce **G**uided **P**ivotal **O**ptimization (GPO), a novel fine-tuning strategy that dives into the reasoning process to enable more effective improvements. GPO first identifies the 'critical step' within a reasoning trajectory - a point that the model must carefully proceed to succeed at the problem. We locate the critical step by estimating the advantage function. GPO then resets the policy to the critical step, samples the new rollout, and prioritizes the learning process on those rollouts. This focus allows the model to learn more effectively from pivotal moments within the reasoning process to improve the reasoning performance. We demonstrate that GPO is a general strategy that can be integrated with various optimization methods to improve reasoning performance. Besides theoretical analysis, our experiments across challenging reasoning benchmarks show that GPO can consistently and significantly enhance the performance of existing optimization methods, showcasing its effectiveness and generalizability in improving LLM reasoning by concentrating on pivotal moments within the generation process.

## 1 Introduction

Large Language Models (LLMs) have demonstrated remarkable capabilities across a wide range of tasks, such as code generation [1], navigating web pages [2], and question answering [3]. However, achieving reliable multi-step reasoning remains a significant frontier in the LLM research [4, 5].

---

[*]Work done while at Northwestern University.

[†]Correspondence to: Xian Wu <xianwu123@meta.com>, Xinyu Xing <xinyu.xing@northwestern.edu>.

39th Conference on Neural Information Processing Systems (NeurIPS 2025).

Complex problem-solving tasks, such as mathematical proofs and editing large codebases, often require generating coherent and logically sound sequences of intermediate steps, forming a reasoning trajectory. While LLMs can produce fluent text, ensuring the correctness of these multi-step reasoning trajectories is challenging, as subtle errors introduced at intermediate steps can lead to the failure of the entire reasoning process [6].

Current state-of-the-art methods for aligning LLMs with desired behaviors, including complex reasoning, often rely on reward modeling fine-tuning techniques [7] like Proximal Policy Optimization (PPO) [8], or preference-based methods like Direct Preference Optimization (DPO) [9]. While effective, these methods typically optimize the model based on preferences or rewards over entire generated trajectories. However, LLMs are prone to making mistakes at intermediate steps, which can lead to the failure of the final answer. These fine-tuning methods are not able to effectively pinpoint and focus on these steps to learn how to handle these points.

In this paper, we propose GPO: **G**uided **P**ivotal **O**ptimization, a novel fine-tuning strategy designed to improve LLM multi-step reasoning capabilities by **Focusing on Pivotal Moments**. Instead of treating reasoning trajectories as a whole, GPO breaks down the process to focus on key moments that are crucial for problem-solving. It first *identifies* the 'critical step' from the reasoning trajectory generated by the LLM. These critical steps are pivotal moments where the model must proceed with precision so as to succeed at the problem, and thus, the model should give special emphasis to those steps. We identify the critical step by estimating the advantage function of each step.

Second, GPO *resets* the trajectory at the critical step and generates a new trajectory by continuing from the critical step. The intuition is that by focusing the learning process on trajectories after these crucial moments, we can more effectively teach the model to navigate challenging reasoning pathways and improve performance. Note that GPO is a general framework that can be integrated into existing fine-tuning methods.

While GPO is as simple to implement for most existing fine-tuning optimization algorithms, we provide a theoretical analysis of the GPO for both online learning and offline preference learning settings. Empirically, we run GPO on diverse reasoning tasks and existing fine-tuning algorithms to show the effectiveness and generalizability of GPO.

Specifically, we make the following key contributions in this paper:

- **Proposal of** GPO: We introduce GPO, a novel fine-tuning strategy that improves LLM reasoning by identifying critical steps in trajectories and prioritizing learning from these pivotal moments to improve the reasoning performance.
- **Theoretical Analysis**: Under natural assumptions, we provide a theoretical analysis of the GPO for the regret bound in the online learning setting, and prove that GPO can be interpreted as a form of advantage-weighted RL in the offline preference learning setting.
- **Empirical Validation**: We conduct extensive experiments on 7 diverse datasets, including general reasoning, mathematical problem solving, and STEM tasks, across 5 different fine-tuning algorithms to validate the effectiveness of GPO.
- **Observation**: We observe that by strategically focusing on learning through critical points, GPO offers a more targeted and effective learning strategy towards enhancing the complex reasoning capabilities of LLMs across diverse optimization frameworks.

To improve transparency and inspire future research, we also release the code and data[3] to facilitate reproducibility and further research.

## 2   Related Work

Since our work aims to improve the reasoning capabilities by identifying critical steps, our work is related to research in reasoning with LLMs, post-training techniques to enhance LLM reasoning, and methods for identifying critical steps in RL.

**LLM Reasoning.** The foundation of LLM reasoning is Chain-of-Thought (CoT) [10], where models are prompted to generate intermediate step-by-step reasoning before the final answer, which can

---

[3]https://github.com/sherdencooper/GPO

boost the performance on complex reasoning tasks. It aligns with how humans reason, where we break down the problem into smaller steps and reason about each step before arriving at the final answer. Subsequent works follow this direction to work on prompting strategies to enhance reasoning [11, 12, 13]. Beyond prompting, there is a growing trend towards developing and fine-tuning LLMs specifically optimized for complex tasks with multi-step reasoning processes. Models like OpenAI O1 [14] or DeepSeek R1 [15] are examples of such models fine-tuned with high-quality reasoning trajectories to achieve state-of-the-art performance. Even without explicit CoT prompting, these models are able to generate step-by-step reasoning trajectories. Our work falls into this category by providing a fine-tuning strategy to improve the multi-step reliability of reasoning trajectories.

**Post-training LLMs for Reasoning.** To supervised fine-tune LLMs for reasoning tasks, it typically requires high-quality annotated datasets [16, 17, 18]. However, the annotation costs of this approach can be significant. To reduce the cost, one method is to synthesize high-quality data from LLMs. One approach uses stronger "teacher" LLMs (*e.g.,* GPT-4o, Gemini) to generate reasoning demonstrations [19, 15, 20, 21]. However, the cost of these strong LLMs especially for those commercial LLMs is still high, and recent work also reveals that the fine-tuning performance may be suboptimal due to the large capacity gap between the teacher and the student LLMs [22, 23, 24, 25, 26]. Thus, recent work starts to explore the LLM self-improvement, where models learn from their own generated data. These works often include methods that filter or refine self-generated samples based on feedback or heuristics [27, 28, 29, 30, 31, 6] or employ advanced prompting techniques during data generation [32, 33, 32]. Models will be trained on the refined samples to improve themselves. Another line of approach, different from data synthesis, is online RL learning [15, 34, 35, 36, 37], where the model interacts with the environment to learn the optimal policy guided by the reward function.

Among the above works, one closely related work to ours for self-improvement is Satori [6], which employs a strategy of randomly resetting the reasoning process at various points and then exploring alternative paths from those reset points to improve the quality of the self-generated data. While similar in use of reset strategy, GPO differs significantly by identifying the critical steps. Besides, as we will demonstrate in §6.2, this random reset strategy is not optimal compared with GPO. Furthermore, Satori only focuses on the offline method without accompanying theoretical analysis, while our method is suitable for both online and offline RL methods, and both provide a theoretical analysis.

**Critical Step Identification.** The concept of identifying critical steps within a sequence is not new in Explainable Reinforcement Learning(XRL), where understanding agent behavior often involves pinpointing critical states or decisions in a trajectory. Various methods have been developed and can be categorized into two types: model-based and model-free methods. For model-based models, they often train a local model to predict the important steps within a trajectory [38, 39, 40, 41]. For model-free methods, they often rely on value functions to identify the critical steps [42, 43, 44]. Our method aligns with model-free techniques. However, directly applying traditional XRL methods to LLM reasoning is challenging because treating generating a single token as an action lacks the semantic meaning in a reasoning process. In §4, we will detail how GPO adapts the core ideas of critical step identification from XRL to the context of multi-step reasoning in LLMs.

## 3 Preliminaries

**Markov decision process and problem setting.** In this work, we consider a **finite-horizon episodic** Markov Decision Process (MDP) defined as $\mathcal{M} = (\mathcal{S}, \mathcal{A}, \{\mathcal{P}\}_h, \{r\}_h, H, d_0)$, where $\mathcal{S}$ is the state space, $\mathcal{A}$ is the action space, $H$ is the episode length, $\{\mathcal{P}\}_h$ denotes the transition dynamics, $\{r\}_h$ is the reward function, and $d_0$ denotes the distribution of initial state. Given a policy $\pi$, the agent seeks to maximize the expected cumulative reward, expressed as $\mathbb{E}_{s \sim d_0} [V_0^\pi(s)]$, where the value function is defined as $V_0^\pi(s) = \mathbb{E}\left[\sum_{h=0}^{H-1} r_h(s_h, a_h) \,\middle|\, s_0 = s, \, a_h \sim \pi_h(\cdot \mid s_h)\right]$, and $r_h(s_h, a_h)$ denotes the reward at step $h$ under state–action pair $(s_h, a_h)$.

In our setting, $d_0$ denotes the distribution of prompt $s_0 = x$. Given a reasoning problem $x \sim d_0$, the goal is to improve a base policy $\pi_{\text{ref}}$ into a refined policy $\pi$ that maximize the expected reward over generated responses $y \sim \pi(\cdot \mid x)$, where $y = (y_0, y_1, \ldots, y_{H-1}) \in \mathcal{Y}$ represents a sequence of reasoning steps (up to $H$), typically separated by newlines. Importantly, rather than treating each generated token as a step, we define each reasoning segment as a step. Since generation is auto-regressive, each step can be interpreted as an action taken by the agent in an MDP with deterministic

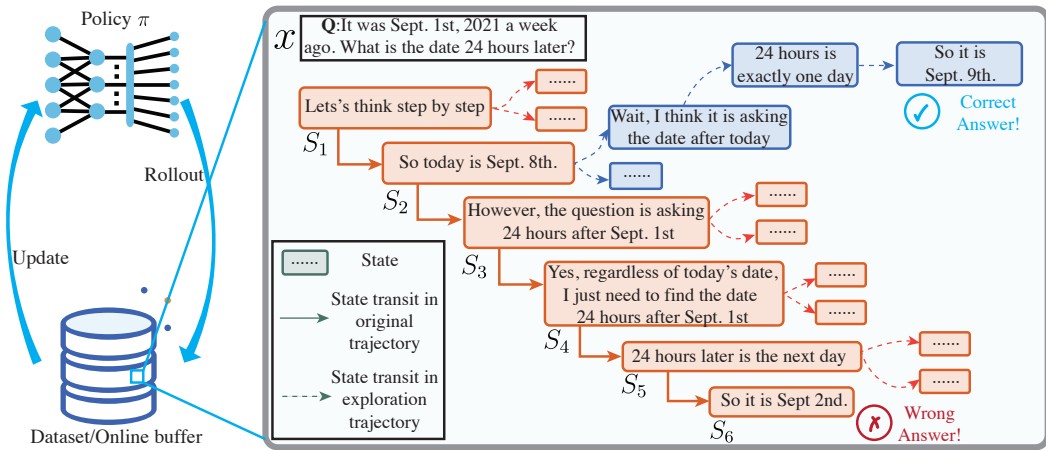

Figure 1: **Overview of our method.** Given an initial trajectory generated by the policy $\pi$ for a reasoning task, GPO segments the trajectory into steps. It then identifies the most critical step via the MC simulation and resets the policy to the critical step to generate a new trajectory. The new trajectory is then added to the dataset or online buffer.

transitions. Specifically, we treat the prefix $(x, y_0, \ldots, y_{h-1})$ as the current state $s_h$, and the next reasoning step $y_h \sim \pi(\cdot \mid s_h)$ as the action taken at $s_h$, resulting in the next state $s_{h+1}$.

**Online policy learning and preference optimization.** For standard MDPs with known reward functions, a variety of online policy gradient algorithms – including Proximal Policy Optimization (PPO) [8] and its recent variant Group Relative Policy Optimization (GRPO) [34]–have been proposed to iteratively improve a policy through direct interactions with environments. These methods have also been extensively applied in LLM training to solve complex mathematical or coding tasks [34, 15, 45], where a clear binary reward function can be easily defined by comparing the LLM-generated output with gold standard solutions. Formally, PPO optimizes the following objective function:

$$\mathcal{J}_{\text{PPO}}(\theta) = \mathbb{E}_{x \sim s_0, y \sim \pi_{\theta_{\text{old}}}(\cdot|x)} \left[ \frac{1}{|y|} \sum_{i=0}^{|y|-1} \min \left( \frac{\pi_\theta(y_i|x, y_{<i})}{\pi_{\theta_{\text{old}}}(y_i|x, y_{<i})} A_i, \text{clip} \left( \frac{\pi_\theta(y_i|x, y_{<i})}{\pi_{\theta_{\text{old}}}(y_i|x, y_{<i})}, 1-\varepsilon, 1+\varepsilon \right) A_i \right) \right] \quad (1)$$

where $\pi_\theta$ and $\pi_{\theta_{old}}$ are the current and old policy models, and $x, y$ are questions and outputs sampled from the question dataset and the old policy $\pi_{\theta_{old}}$, respectively. $\varepsilon$ is a clipping-related hyper-parameter introduced in PPO for stabilizing training. $A_i$ is the advantage function.

The MDP formulation of preference learning was recently explored in [9, 46, 47]. In this setting, the true reward function is typically unobservable; instead, we are given an offline dataset of trajectory pairs $\mathcal{D} = \{(x, y^+, y^-)\}$ labeled with human preferences. Prior approaches in reinforcement learning from human feedback (RLHF) [48, 7] typically follow a two-stage pipeline: (1) learning a reward function from preference data via the Bradley-Terry(BT) model [49], and (2) training a policy via the PPO algorithm to maximize the learned reward. In contrast, [9] establishes a direct connection between the optimal policy $\pi^*$ and its associated reward function, and proposes a surrogate objective, referred to as Direct Preference Optimization (DPO) to directly learn the optimal policy from the offline preference pairs:

$$\min_\pi \; \mathcal{L}_{\text{DPO}}(\pi) := \mathbb{E}_{(x,y^+,y^-))\sim\mathcal{D}} \left[ \sigma \left( \beta \log \frac{\pi(y^+ \mid x)}{\pi_{\text{ref}}(y^+ \mid x)} - \beta \log \frac{\pi(y^- \mid x)}{\pi_{\text{ref}}(y^- \mid x)} \right) \right], \quad (2)$$

where $\sigma(\cdot)$ denotes the sigmoid function, $\beta$ is a temperature parameter, and $\pi_{\text{ref}}$ is a fixed reference policy. There are variants of DPO that use different loss functions like SimPO [47] and ORPO [46], or eliminate the need for paired samples, like KTO [46]. We introduce them in detail in Appendix D.

## 4 Method

At a high level, given a reasoning trajectory $y$ generated by current policy $\pi$ (*e.g.*, $\pi = \pi_{\text{ref}}$), we identify the most critical reasoning step $y_h$ and refine $\pi$ from this point using either an on-policy

---
**Algorithm 1:** GPO Optimization Framework

---
1: **Procedure-I: Online Policy Training (PPO-based)**
2: **Input:** Initial LLM policy $\pi^0 = \pi_{\text{ref}}$, reasoning dataset $D_r$
3: **for** iteration = 1, 2, ..., $T$ **do**
4:    $\mathcal{D} \leftarrow \emptyset$
5:    **for** n = 1, 2, ... **do**
6:       Random sampling a question $x$ from the reasoning dataset $D_r$
7:       Run $\pi^t$ to generate a $K$-step reasoning trajectory $y = (y_0, \ldots, y_{K-1})$ split by newlines
8:       Identify the critical step $y_m$ with maximal advantage $A^{\pi^t}(x, y_{0:i-1}; y_i)$
9:       Reset $\pi^t$ to $y_m$ and roll-out $\pi^t$ to generate trajectory $y' = (y_m, \ldots, y'_{K-1})$
10:       Add trajectory $y'$ and the final reward $r$ to $\mathcal{D}$
11:    **end for**
12:    Optimize $\pi^t$ with respect to the policy gradient loss (*e.g.,* PPO loss) in Eqn. 1 on $\mathcal{D}$
13: **end for**

---
14: **Procedure-II: Preference Data Generation and Optimization (DPO-based)**
15: **Input:** Supervised-finetuned base policy $\pi_{\text{ref}}$, reasoning dataset $D_r$
16: $\mathcal{D} \leftarrow \emptyset$
17: **for** iteration = 1, 2, ..., $T$ **do**
18:    Repeat the sampling, trajectory generation using $\pi_{\text{ref}}$, and critical step identification as in **Procedure-I** to extract the important step $y_m$ from trajectory $y$.
19:    Generate two continuations starting from $y_m$ to obtain a positive trajectory $y^+ = (y_0, \ldots, y_m, \ldots, y^+_{K-1})$ and a negative trajectory $y^- = (y_0, \ldots, y_m, \ldots, y^-_{K-1})$
20:    Add the preference pair $(x, y^+, y^-)$ to $\mathcal{D}$
21: **end for**
22: Optimize $\pi$ with respect to the preference loss (*e.g.,* DPO loss) in Eqn. 2 on $\mathcal{D}$

---

algorithm such as PPO or an offline preference method such as DPO. Intuitively, revisiting these pivotal steps enables exploration of potential alternative reasoning paths, overcoming the training bottlenecks of current policy. To illustrate the core mechanism of GPO, consider the example shown in Figure 1. The task is *It was Sept. 1st, 2021 a week ago. What is the date 24 hours later?*. An initial trajectory sampled from the policy $\pi$ generates a multi-step reasoning process towards the question; however, it misinterprets the question and gives the wrong answer. GPO first segments this trajectory into multiple reasoning steps (*e.g.,* $S_1, \ldots S_6$). Here, the state is the sequence of all previous reasoning steps, and the action is the next reasoning step to be taken. Then, it identifies the most critical reasoning step via Monte Carlo (MC) estimation of the advantage function from RL. $S_2$ is identified as the most critical, as the alternative continuation after $S_2$ could yield a correct final answer, while the continuation after other steps cannot during the MC simulation. Next, it resets the trajectory at $S_2$ and generates a new trajectory $y'$ by continuing from $S_2$ with the current policy $\pi$, then adds the trajectory $y'$ to the dataset or online buffer. By focusing on trajectories associated with the critical step, GPO directs the learning process towards the specific reasoning step where the policy should focus, thereby enhancing the reasoning performance of the policy.

**Identifying the Critical Step via Advantage.** We measure the importance of each reasoning step using advantage functions in RL. For any candidate step $y_i$ within a predicted reasoning trajectory $y$, its advantage quantifies the incremental value of taking that step, defined as the relative change in $Q$-value when adding $y_i$ to the current partial sequence $y_{0:i-1}$, *i.e.,* $A^\pi(x, y_{0:i-1}; y_i) = Q^\pi(x, y_{0:i-1}; y_i) - Q^\pi(x, y_{0:i-2}; y_{i-1})$[4] . Here, the $Q$-function $Q^\pi(x, y_{0:i-1}; y_i)$ estimates the expected future reward of taking action $y_i$ after observing prefix $y_{0:i-1}$ under an auxiliary policy $\pi$. Formally, given a problem $x$ with the gold answer $y_{gold}$, and a predicted trajectory $y$ sampled from the policy $\pi$, we define: $Q^\pi(x, y_{0:i}; y_i) = \mathbb{E}_{y^{\text{new}}_{i+1:H-1} \sim \pi(\cdot|x, y_{1:i})} \left[ r\left([y_{0:i}, y^{\text{new}}_{i+1:H-1}], y_{gold}\right)\right]$. The reward function $r(\cdot)$ compares the completed trajectory (including sampled future steps) with the ground-truth solution $y_{gold}$. The policy $\pi$ governs how future steps are sampled; It can be unbiasedly

---
[4]In RL, the advantage function is defined as $A(s_t, a_t) = Q(s_t, a_t) - V(s_t)$. In our setting, with deterministic transitions and zero intermediate rewards, this expression simplifies to the difference between consecutive $Q$-values: $Q(s_t, a_t) - Q(s_{t-1}, a_{t-1})$.

estimated via MC simulations [50] by sampling multiple continuations from the current step under policy $\pi$. In §6.3, we will evaluate how the number of MC simulations affects the performance of GPO.

**Fine-Tuning with PPO or DPO.** Once the most critical reasoning step in a trajectory is identified—formally, the step with the highest advantage—we refine the policy by exploring alternative continuations from this step. Instead of treating the entire trajectory uniformly, we reset the policy to a critical step and sample new rollouts conditioned on it. As we will demonstrate in §5, this advantage-weighted-style sampling strategy reduces the regret of the final converged policy and enables more efficient online policy improvement. The resulting high-quality trajectories are then incorporated into the training set to guide policy updates. Our framework supports two complementary optimization methods: (i) online policy-gradient optimization like PPO, which updates the policy based on reward feedback (**Procedure-I** in Algorithm 1) and (ii) offline preference optimization like DPO, which leverages pairwise preferences (**Procedure-II** in Algorithm 1).

## 5 Theoretical Analysis

In this section, we present theoretical results and insights related to Algorithm 1.

### 5.1 Online policy gradient algorithm

Before proceeding, we generalize **Procedure-I** in Algorithm 1 by sampling the critical step with probability proportional to $e^{\gamma A^{\pi^t}(s,a)}$, where $\gamma > 0$ is a temperature value. The original algorithm can be viewed as the limiting behavior for a sufficiently large $\gamma$. Since we operate in a finite-horizon episodic MDP, we evaluate the performance of the online policy gradient algorithm via its regret: $Regret = \frac{1}{T}\sum_{t=1}^{T}\left(V_0^{\pi^*}(s_0) - V_0^{\pi^t}(s_0)\right)$. We begin by stating the following assumption regarding the $Q$-function.

**Assumption 5.1** (Bounded $Q$-value). *Suppose we have a function class $\mathcal{F}$ and $Q_h^{\pi^t} \in \mathcal{F}$ holds for the Q function of policy $\pi^t$, $\forall t = 1, 2, \ldots, T$. We assume that $0 \leq Q_h^{\pi^t}(s_h, a_h) \leq r_{\max}$ for all $Q_h^{\pi^t}(s_h, a_h) \in \mathcal{F}$, $s_h \in \mathcal{S}$, $a_h \in \mathcal{A}$.*

Assumption 5.1 is reasonable because reasoning tasks typically involve a bounded final reward . We further present Theorem 5.2 to bound the regret for the online policy gradient algorithm.

**Theorem 5.2.** *Under Assumption 5.1, with probability $1 - \delta$, we have the following regret bound:*

$$\frac{1}{T}\sum_{t=1}^{T}\left(V_0^{\pi^*}(s_0) - V_0^{\pi^t}(s_0)\right) \leq r_{max}H\sqrt{\frac{T\log|\mathcal{A}|}{2}} + CTHr_{\max}^2\log(\frac{TH|\mathcal{F}|}{\delta})\sqrt{w_{max}(\gamma)} \quad (3)$$

where $C$ is a constant and $w_{max}(\gamma)$ represents the step-wise concentrability [51] between the optimal policy $\pi^*$ and our policy. We also show that an increasing $\gamma$ will tighten the regret bound in §C.1, which validates the importance of our advantage reweighting technique.

### 5.2 Preference optimization

For the DPO-based optimization in **Procedure-II** of Algorithm 1, we consider a conceptual variant inspired by per-step DPO [52, 21], which introduces preference comparisons at each individual reasoning step, in contrast to the standard DPO that operates over full trajectories. This step-wise modeling leads to the following result:

**Theorem 5.3.** *Let $\mathcal{D}$ consist of tuples $(x, [y_{0:i-1}, y_i^+], [y_{0:i-1}, y_i^-])$, where $y_{0:i-1} \sim \pi_{ref}$ and both continuations $y_i^{\pm}$ are drawn from $\pi_{ref}(\cdot \mid x, y_{0:i-1})$, with preferences determined by the advantage $A^{\pi_{ref}}(x, y_{0:i-1}; \cdot)$. Then, the optimal policy from minimizing Eqn. 2 coincides with the solution to the following advantage-weighted RL objective:*

$$\max_{\pi} \mathbb{E}_{x\sim d_0,\ y\sim\pi_{ref}(\cdot|x)}\left[\sum_{i=0}^{H-1}\log\pi(y_i \mid x, y_{0:i-1}) \cdot \exp\left(\frac{A^{\pi_{ref}}(x, y_{0:i-1}; y_i)}{\beta}\right)\right] \quad (4)$$

Table 1: **Comparison of GPO-enhanced methods against baselines.** The better results between GPO-enhanced and baseline methods are highlighted in bold. Each training result is the average of 3 runs with different random seeds.

| Algorithms | Test Accuracy (%) | | | | | | |
|---|---|---|---|---|---|---|---|
| | BBH | MATH | GSM8K | MMLU | MMLUPro | AIME-2024 | AIME-2025 |
| Base Model | 59.97 | 71.60 | 86.50 | 54.09 | 38.80 | 13.33 | 16.67 |
| PPO | 61.82 | 79.60 | 86.96 | 56.66 | 47.47 | 26.67 | 23.33 |
| GPO-PPO | **63.48** | **87.80** | **87.44** | **59.39** | **51.05** | **30.00** | **26.67** |
| DPO | 63.20 | 82.40 | 86.05 | 57.08 | 48.28 | 20.00 | 20.00 |
| GPO-DPO | **64.25** | **86.80** | **88.48** | **58.93** | **51.93** | **26.67** | **26.67** |
| KTO | 62.86 | 77.20 | 89.31 | 59.42 | 49.02 | 20.00 | 20.00 |
| GPO-KTO | **64.31** | **79.60** | **90.25** | **61.35** | **50.52** | **23.33** | **26.67** |
| SimPO | 61.97 | 72.20 | 86.58 | 56.93 | 45.70 | 20.00 | 23.33 |
| GPO-SimPO | **62.58** | **74.00** | **88.35** | **57.44** | **47.74** | **23.33** | **26.67** |
| ORPO | 61.75 | 75.20 | 87.26 | 57.72 | 46.66 | 20.00 | 20.00 |
| GPO-ORPO | **62.28** | **78.20** | **88.17** | **58.72** | **48.65** | **23.33** | **23.33** |

In summary, Theorem 5.3 shows that per-step DPO, with preference determined by the advantage function, corresponds to advantage-weighted RL, where the advantage-based preferences implicitly reweight the log-likelihood at each step. Compared to standard offline supervised fine-tuning (SFT), this reweighting allows the model to focus more on critical decision points, leading to more targeted updates and enhanced overall performance [53, 54]. We provide the proof details in Appendix C.2.

# 6 Experiments

**Implementation Details.** We primarily employ the DeepSeek-R1-Distill-Qwen-7B model as the base model for our experiments, selected for its instruction-following and reasoning capabilities, as well as training efficiency. To further enhance data quality and reduce training costs, we follow prior works [19, 55, 6] to filter the data based on question difficulty. For advantage function estimation, we use 4 MC samples for each step. Additional implementation details are available in §E.2.

**Baseline Methods.** To evaluate the enhancement provided by GPO, we compare the performance of several established fine-tuning algorithms against their GPO-enhanced counterparts. The baseline methods include online RL method PPO, and preference-based algorithms DPO, KTO, SimPO, and ORPO. Here, we only consider one online RL method because PPO is one of the most popular online RL methods widely used in the community, and the computation resources needed for online RL are much more significant than those for offline RL methods. Consistent with prior work on reasoning meibitasks [15, 55, 19], we utilize a rule-based reward function for the PPO implementation. For a fair comparison, identical hyperparameters are used for each baseline method and its corresponding GPO-enhanced version. We use the LoRA method [56] for fine-tuning. Detailed hyperparameter configurations are provided in §E.3.

**Dataset and Evaluation Metrics.** We evaluate the effectiveness of our method on 7 diverse datasets covering a range of reasoning tasks. For mathematical problem solving, we use GSM8K [57], MATH-500 [58], AIME-2024 [59], and AIME-2025 [60]. For general reasoning, we utilize BIG-Bench Hard (BBH) [61]. For STEM problem solving, we employ MMLU [62] and MMLUPro [63]. Standard train/test splits are used for GSM8K, MATH, MMLU, and MMLUPro. Following prior work [19, 64, 65], the AIME training set consists of problems from 1983-2023. For BBH, we randomly split the dataset into training set and test set by sub-task. Further dataset statistics can be found in §E.1. Accuracy is evaluated using zero-shot pass@1 accuracy via greedy decoding. We use different random seeds for training and report the average performance over 3 runs.

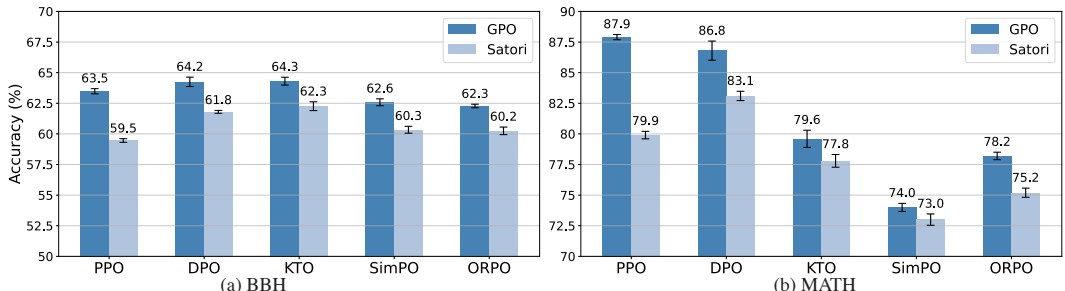

Figure 2: **Ablation study results on BBH and MATH.** We compare the performance of the standard GPO method and Satori's strategy that randomly identifies the critical step in the trajectory. Each bar represents the average performance of 3 runs, with error bars indicating the standard deviation.

## 6.1 Main Results

The results are presented in Table 1. The table clearly demonstrates the effectiveness and generalizability of GPO. Across all 7 datasets and all 5 optimization algorithms, integrating GPO consistently leads to improved test accuracy compared to the respective baseline method. This consistent improvement underscores the robustness of leveraging critical step identification to enhance fine-tuning.

Notably, the performance gains achieved by GPO are often substantial. For instance, GPO-PPO and GPO-DPO show significant accuracy increases on the MATH dataset compared to standard PPO and DPO. Similar positive trends are observed across other datasets like MMLUPro and the AIME benchmarks. While the magnitude of improvement varies depending on the specific dataset and baseline algorithm, the consistent improvement validates our core hypothesis: focusing learning on critical reasoning steps provides a more effective training signal, leading to enhanced reasoning capabilities in the fine-tuned models.

## 6.2 Ablation Study

To better understand how learning from the critical steps contributes to performance improvement, we conduct an ablation study on the BBH and MATH datasets to analyze the impact of critical step identification. Satori [6] has shown that randomly locating one step in the trajectory, then resetting and exploring the trajectory from that step, could help augment the training data and improve the performance of RLHF. Inspired by this, we randomly locate the critical step in the trajectory and compare the performance of the standard GPO method. The results are presented in Figure 2.

The GPO method consistently outperforms Satori's random selection baseline across both datasets. The difference is particularly obvious on MATH, where PPO with GPO achieves 87.9% accuracy, significantly higher than the 79.9% achieved when using Satori's strategy. These findings suggest that the performance gains of GPO are not merely due to the random resetting mechanism itself. Instead, the identification and learning from critical steps, leading to a more effective training signal for crucial points, is a key factor driving the improvement observed during training.

## 6.3 Monte Carlo Simulation and Model Size Scaling

To understand the scalability of GPO, we investigate its performance to two key factors: the number of MC simulations used for critical step identification and the size of the base model. First, we vary the number of MC simulations from 2 to 16. Second, we apply GPO to models of varying scales, specifically the DeepSeek-R1-Distill-Qwen series (1.5B, 7B, 14B, 32B parameters) and the DeepSeek-R1-Distill-Llama-70B model. These experiments are conducted on the MATH dataset, using both DPO and KTO as the underlying optimization algorithms. The results are presented in Figure 3.

The illustration reveals that increasing the number of MC simulations generally improves the performance benefit of GPO. This suggests that a more accurate estimation of the advantage function, derived from more simulation samples, can lead to better training performance. However, the perfor-

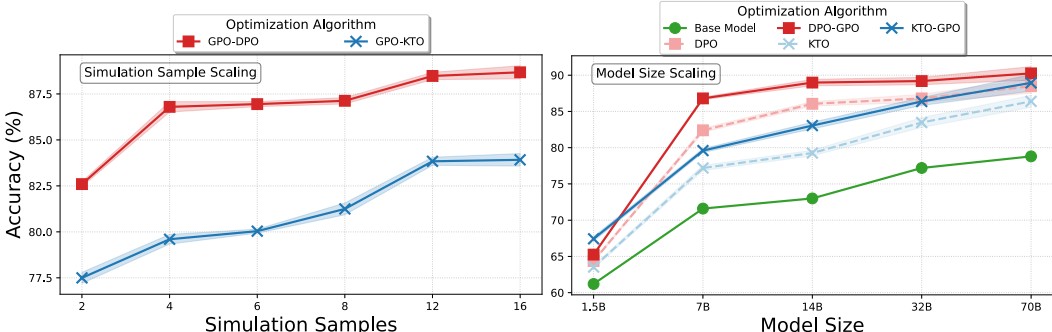

Figure 3: **Scaling behavior of** GPO. Performance impact of varying number of MC samples (left) and applying GPO across different model sizes (right) on MATH using DPO/KTO.

mance gains appear to saturate beyond 12 simulation samples, potentially because the estimation of the advantage function converges. It suggests that the overhead of GPO can be reduced by achieving a balance between the number of simulation samples and the performance gain.

Furthermore, the results also demonstrate that GPO consistently outperforms the corresponding baseline optimization algorithms (DPO/KTO without GPO) across all tested model sizes, from 1.5B to 70B parameters. This consistent improvement highlights the robustness of the GPO and its applicability to larger, more capable language models.

## 7 User Study

We explore whether GPO's identification of 'critical steps' aligns with humans. We conduct a user study to evaluate the correlation between steps identified by GPO and those by human evaluators.

**Study Setup.** We designed a study involving 50 participants, recruited from college students. Participants are presented with five reasoning problems, each accompanied by a corresponding reasoning trajectory generated by the base LLM. For each trajectory, participants are asked to identify the single step they believed was the most critical point during the reasoning process for the final answer. They are presented with four optional steps for each problem: one step identified by GPO, while the other three options are randomly selected. Further details regarding the specific questions and trajectories are provided in Appendix F.

**Results and Discussion.** The results indicate a strong alignment between the critical steps identified by GPO and human judgment. Across the five questions evaluated, the percentage of participants who selected the GPO-identified step as the most critical was 44%, 68%, 88%, 76%, and 56%, respectively. This high degree of agreement suggests that the steps pinpointed by our process are also often recognized by humans as the crucial points. These findings provide qualitative validation for the core mechanism of GPO, supporting the hypothesis that its empirical improvements come from.

## 8 Discussion and Limitations

While GPO shows promise in enhancing LLM reasoning, we acknowledge several limitations and areas for future work.

A key limitation is the computational overhead from the Monte Carlo estimation of future returns at each step. This adds a non-negligible cost, increasing PPO training time by approximately 1.9x and offline data preparation by 1.8x, a common challenge for related methods [21, 66, 32]. However, this trade-off is manageable. First, as shown in §6.3, performance gains saturate after a certain number of simulations, allowing for a practical balance between accuracy and cost. Second, to adapt GPO for very long trajectories, we employ a simple heuristic of grouping multiple generation steps into a single logical step. We validated this approach on a challenging long-context reasoning subset of BigBenchExtremeHard (BBEH) [67], where this simple grouping strategy enabled GPO to achieve 36.0% accuracy—a substantial +7% improvement over the DPO baseline. This result confirms that GPO can be effectively scaled to complex tasks, and we provide the experiment details in §E.5.

Furthermore, there are several promising avenues for future research. To further enhance efficiency in the online setting, the current Monte Carlo estimation could be replaced with a more sample-efficient alternative like Generalized Advantage Estimation (GAE)[8], leveraging the value network trained by the PPO algorithm. Beyond efficiency, open questions remain regarding the identification of critical steps. For instance, could model-based explainability techniques[38, 39] offer higher fidelity compared to our model-free approach? Investigating hybrid methods, such as using powerful commercial LLMs like GPT-4o to assist in identifying critical steps, could also be a fruitful direction.

Finally, evaluating the quality of identified critical steps currently relies on downstream task performance and expensive, hard-to-scale human judgment. Benchmarks like ProcessBench [68] only focus on identifying first incorrect step in a failed trajectory, which is not fully aligned with the scope of critical step identification. Developing automated metrics or benchmarks to reliably assess critical step quality and relevance would significantly benefit future research and accelerate iteration on methods like GPO. We hope our work encourages further exploration of these important questions.

## 9 Conclusion

In this work, we introduce GPO, a strategy to enhance LLM reasoning by identifying critical steps within generation trajectories. Supported by theoretical guarantees, GPO demonstrably boosted performance across seven diverse reasoning datasets when integrated with five optimization algorithms, highlighting its generalizability. Furthermore, the user study confirms that the critical steps identified by GPO align well with human judgments of pivotal moments in reasoning failures. We believe GPO represents a valuable step towards more robust and reliable reasoning in LLMs, and we hope it inspires further research into targeted trajectory optimization and analysis.

## Acknowledgement

This work was supported in part by NSF Grants 2225234 and 2225225. This research was also supported in part through the computational resources and staff contributions provided for the Quest high performance computing facility at Northwestern University which is jointly supported by the Office of the Provost, the Office for Research, and Northwestern University Information Technology. We thank the anonymous reviewers for their constructive feedback and valuable suggestions that helped improve this work.

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

## A    Broader Impact

Beyond enhancing the reasoning capabilities of LLMs, GPO holds significant potential for advancing the trustworthiness and transparency of these models. By employing our method, we can identify and highlight the critical steps within the reasoning processes of LLMs. This capability not only aids in demystifying the decision-making pathways of these models but also empowers users to gain a deeper understanding of how conclusions are reached. Consequently, this increased clarity can foster greater trust in the outputs generated by LLMs. Furthermore, by making the reasoning process more transparent, stakeholders can more easily verify and validate the model's decisions, thereby enhancing the overall reliability and acceptance of LLMs in various applications. There could be potential negative societal impacts of our work. For example, if the critical steps are not properly highlighted, it could lead to misinformation and harm the trust in LLMs. However, we believe that the potential benefits of our work outweigh the potential risks, as we open-source our method and inspire more research on the critical steps identification in LLMs.

## B    Ethics Considerations

Our work involves the use of a user study to evaluate whether the identified critical steps align with human preferences. We have detailed the instructions we gave to participants in Appendix F to make it transparent and reproducible. The questions we designed only evaluate the critical steps; thus, they pose no harm to the participants. We **consult the IRB office at our institution and receive an exemption for this study**. Moreover, we do not collect any sensitive information from the users.

## C    Theory

### C.1    Online policy gradient algorithm

First, we introduce the performance difference lemma from [36].

**Lemma 1** (performance difference lemma [36])**.** *For any policy $\pi, \pi'$ and reward function $r$, we have*

$$V_0^\pi(s_0) - V_0^{\pi'}(s_0) = \sum_{h=0}^{H-1} \mathbb{E}_{s_h \sim d_h^\pi}[\langle Q_h^{\pi'}(s_h), \pi_h(s_h) - \pi'_h(s_h)\rangle] \tag{5}$$

Based on the performance difference lemma, we can rewrite the regret as

$$regret = \sum_{t=1}^{T}(V_0^{\pi^*}(s_0) - V_0^{\pi^t}(s_0)) = \sum_{t=1}^{T}\sum_{h=0}^{H-1}\mathbb{E}_{s_h \sim d_h^{\pi^*}}[\langle Q_h^{\pi^t}(s_h), \pi_h^*(s_h) - \pi_h^t(s_h)\rangle] \quad (6)$$

where $\hat{Q}_h^{\pi^t}(.)$ is the estimated Q function for the policy $\pi^t$ and $d_h^{\pi^*}$ denotes the state-action visitation measure given the optimal policy $\pi^*$. So we decompose the above equation to be

$$regret = \sum_{t=1}^{T}\sum_{h=0}^{H-1}\mathbb{E}_{s_h \sim d_h^{\pi^*}}[\langle \hat{Q}_h^{\pi^t}(s_h), \pi_h^*(s_h) - \pi_h^t(s_h)\rangle] \quad \text{(term(1))}$$

$$+ \sum_{t=1}^{T}\sum_{h=0}^{H-1}\mathbb{E}_{s_h \sim d_h^{\pi^*}}[\langle Q_h^{\pi^t}(s_h) - \hat{Q}_h^{\pi^t}(s_h), \pi_h^*(s_h) - \pi_h^t(s_h)\rangle] \quad \text{(term(2))}$$

We present the following theorem to bound the term (1).

**Theorem C.1.** *Suppose Assumption 5.1 holds, we have*

$$\sum_{t=1}^{T}\sum_{h=0}^{H-1}\mathbb{E}_{s_h \sim d_h^{\pi^*}}\left[\langle \hat{Q}_h^{\pi^t}(s_h), \pi_h^*(s_h) - \pi_h^t(s_h)\rangle\right] \leq r_{max}H\sqrt{\frac{T\log|\mathcal{A}|}{2}} \quad (7)$$

**Proof of Theorem C.1.** First, we decompose term (1) into two parts.

$$\langle \hat{Q}_h^{\pi^t}(s_h), \pi_h^*(s_h) - \pi_h^t(s_h)\rangle$$
$$=\langle \hat{Q}_h^{\pi^t}(s_h), \pi_h^*(s_h) - \pi_h^{t+1}(s_h)\rangle + \langle \hat{Q}_h^{\pi^t}(s_h), \pi_h^{t+1}(s_h) - \pi_h^t(s_h)\rangle$$
$$\leq\langle \hat{Q}_h^{\pi^t}(s_h), \pi_h^*(s_h) - \pi_h^{t+1}(s_h)\rangle + \|\hat{Q}_h^{\pi^t}(s_h)\|_\infty\|\pi_h^{t+1}(s_h) - \pi_h^t(s_h)\|_1$$

By Assumption 5.1, we have $\|\hat{Q}_h^{\pi^t}(s_h)\|_\infty \leq r_{max}$, which further implies

$$\langle \hat{Q}_h^{\pi^t}(s_h), \pi_h^*(s_h) - \pi_h^t(s_h)\rangle$$
$$\leq\langle \hat{Q}_h^{\pi^t}(s_h), \pi_h^*(s_h) - \pi_h^{t+1}(s_h)\rangle + r_{\max}\|\pi_h^{t+1}(s_h) - \pi_h^t(s_h)\|_1$$

Note that the policy update formula for the online policy gradient algorithm has a closed-form expression [51]

$$\pi_h^{t+1}(s_h) = \frac{1}{Z_h^t(s_h)}\pi_h^t(s_h)e^{\eta\hat{Q}_h^{\pi^t}(s_h)} \quad (8)$$

where $Z_h^t(s_h)$ is a normalization factor. Take the logarithm over both sides, and we have

$$\eta\hat{Q}_h^{\pi^t}(s_h) = \log Z_h^t(s_h) + \log \pi_h^{t+1}(s_h) - \log \pi_h^t(s_h) \quad (9)$$

Thus,

$$\langle \hat{Q}_h^{\pi^t}(s_h), \pi_h^*(s_h) - \pi_h^{t+1}(s_h)\rangle$$
$$=\langle \frac{1}{\eta}(\log Z_h^t(s_h) + \log \pi_h^{t+1}(s_h) - \log \pi_h^t(s_h)), \pi_h^*(s_h) - \pi_h^{t+1}(s_h)\rangle$$

Note that $\log Z_h^t(s_h)\sum_a[\pi^*(a|s_h) - \pi^{t+1}(a|s_h)] = 0$. We can reorganize the above equation as

$$\langle \hat{Q}_h^{\pi^t}(s_h), \pi_h^*(s_h) - \pi_h^{t+1}(s_h)\rangle$$
$$=-\frac{1}{\eta}KL(\pi_h^*(s_h)\|\pi_h^{t+1}(s_h)) + \frac{1}{\eta}KL(\pi_h^*(s_h)\|\pi_h^t(s_h))$$
$$-\frac{1}{\eta}KL(\pi_h^{t+1}(s_h)\|\pi_h^t(s_h))$$

Note that by Pinsker's inequality, we have

$$\|\pi_h^{t+1}(s_h) - \pi_h^t(s_h)\|_1 \le \sqrt{\frac{1}{2}KL(\pi_h^{t+1}(s_h)\|\pi_h^t(s_h))} \tag{10}$$

We further obtain the following bound for $\langle \hat{Q}_h^{\pi^t}(s_h), \pi_h^*(s_h) - \pi_h^{t+1}(s_h)\rangle$:

$$\langle \hat{Q}_h^{\pi^t}(s_h), \pi_h^*(s_h) - \pi_h^{t+1}(s_h)\rangle$$
$$\le -\frac{1}{\eta}KL(\pi_h^*(s_h)\|\pi_h^{t+1}(s_h)) + \frac{1}{\eta}KL(\pi_h^*(s_h)\|\pi_h^t(s_h))$$
$$-\frac{1}{2\eta}\|\pi^{t+1}(s_h) - \pi^t(h)\|_1^2$$

Therefore, we have the following upper bound for $\langle \hat{Q}_h^{\pi^t}(s_h), \pi_h^*(s_h) - \pi_h^t(s_h)\rangle$:

$$\langle \hat{Q}_h^{\pi^t}(s_h), \pi_h^*(s_h) - \pi_h^t(s_h)\rangle$$
$$=\langle \hat{Q}_h^{\pi^t}(s_h), \pi_h^*(s_h) - \pi_h^{t+1}(s_h)\rangle + \langle \hat{Q}_h^{\pi^t}(s_h), \pi_h^{t+1}(s_h) - \pi_h^t(s_h)\rangle$$
$$\le\frac{1}{\eta}[KL(\pi_h^*(s_h)\|\pi_h^t(s_h)) - KL(\pi_h^*(s_h)\|\pi_h^{t+1}(s_h))] + r_{\max}\|\pi_h^{t+1}(s_h) - \pi_h^t(s_h)\|_1$$
$$-\frac{1}{2\eta}\|\pi_h^{t+1}(s_h) - \pi_h^t(s_h)\|_1^2$$
$$\le\frac{1}{\eta}[KL(\pi_h^*(s_h)\|\pi_h^t(s_h)) - KL(\pi_h^*(s_h)\|\pi_h^{t+1}(s_h))] + \frac{1}{2}\eta r_{\max}^2$$

Summarizing over all horizons and over all iterations, we have

$$\sum_{t=1}^T \sum_{h=0}^{H-1} \mathbb{E}_{s_h \sim d_h^{\pi^*}} \langle \hat{Q}_h^{\pi^t}(s_h), \pi_h^*(s_h) - \pi_h^t(s_h)\rangle$$
$$\le \sum_{t=1}^T \sum_{h=0}^{H-1} \mathbb{E}_{s_h \sim d_h^{\pi^*}} \left\{\frac{1}{\eta}[KL(\pi_h^*(s_h)\|\pi_h^t(s_h)) - KL(\pi_h^*(s_h)\|\pi_h^{t+1}(s_h))] + \frac{1}{2}\eta r_{\max}^2\right\}$$
$$= \sum_{h=0}^{H-1} \mathbb{E}_{s_h \sim d_h^{\pi^*}} \left\{\frac{1}{\eta}[KL(\pi_h^*(s_h)\|\pi^1(s_h)) - KL(\pi_h^*(s_h)\|\pi_h^{t+1}(s_h))] + \frac{1}{2}\eta r_{\max}^2 T\right\}$$

Note that initially $\pi^1(a_h|s_h)$ is a uniform distribution over the action space $\mathcal{A}$. We further have

$$\sum_{t=1}^T \sum_{h=0}^{H-1} \mathbb{E}_{s_h \sim d_h^{\pi^*}} \langle \hat{Q}_h^{\pi^t}(s_h), \pi_h^*(s_h) - \pi_h^t(s_h)\rangle$$
$$\le \sum_{h=0}^{H-1} \mathbb{E}_{s_h \sim d_h^{\pi^*}} \left(\frac{\log|\mathcal{A}|}{\eta} + \frac{1}{2}\eta r_{\max}^2 T\right)$$

Let $\eta = \sqrt{\frac{2\log|\mathcal{A}|}{r_{\max}^2 T}}$ and we get

$$\sum_{t=1}^T \sum_{h=0}^{H-1} \mathbb{E}_{s_h \sim d_h^{\pi^*}} \langle \hat{Q}_h^{\pi^t}(s_h), \pi_h^*(s_h) - \pi_h^t(s_h)\rangle$$
$$\le \sum_{h=0}^{H-1} \mathbb{E}_{s_h \sim d_h^{\pi^*}} \left(\frac{\log|\mathcal{A}|}{\eta} + \frac{1}{2}\eta r_{\max}^2 T\right)$$
$$= r_{max} H \sqrt{\frac{T\log|\mathcal{A}|}{2}}$$

**Algorithm 2:** Q function estimation

1: **Input:** Num of rollout $K$, Current policy $\pi^t$, Reward $r$, Hyperparameter $\eta$
2: Initialize: $\mathcal{D}_t = \emptyset$
3: **for** $k = 1, 2, \ldots, K$ **do**
4:     Collect one trajectory $\{(s_0^k, a_0^k, s_1^k, a_1^k, \ldots, s_{H-1}^k, a_{H-1}^k)\}$.
5:     Compute the advantage $A^{\pi^t}(s, a) = Q^{\pi^t}(s, a) - V^{\pi^t}(s)$ for each $(s, a)$.
6:     Sample $(s, a)$ with probability proportional to $\exp(\gamma A(s, a))$. Denote the sampled state-action pair as $(s_m^k, a_m^k)$.
7:     Reset $\pi^t$ to $s_m^k$ and follow $\pi^t$ to generate a trajectory $\{(s_m^k, a_m^k, \ldots, s_H^{k'}, a_H^{k'})\}$.
8:     Compute $q_m^k = \sum_{j=m}^{H-1} r_j$ and add $(s_m^k, a_m^k, y_m^k, q_m^k)$ into $\mathcal{D}_t$.
9: **end for**
10: Compute $\hat{Q}^{\pi^t} = \underset{f}{\arg\min} \mathbb{E}_{\mathcal{D}_t}[(f(s, a) - q)^2]$.

We present Algorithm 2 to obtain the estimate $\hat{Q}^{\pi^t}$ function.

Note that under Algorithm 2, we actually reweight the state-action occupancy, i.e., reweighted state-action occupancy

$$d_h^\rho(s, a) = d_h^{\pi^t}(s, a) e^{\gamma A^{\pi^t}(s,a)} / Z_h(s)$$

where $Z_h(s)$ is a normalization factor

$$Z_h(s) = \sum_{a \in \mathcal{A}} d_h^{\pi^t}(s, a) e^{\gamma A^{\pi^t}(s,a)}$$

Then, we have the following lemma:

**Lemma 2** ([69]). *With probability $1 - \delta$, Algorithm 2 guarantees that, for every $t = 1, 2, \ldots, T$ and $h = 1, 2, \ldots, H$*

$$\mathbb{E}_{(s_h, a_h) \sim d_h^\rho}[\hat{Q}_h^{\pi^t}(s_h, a_h) - Q_h^{\pi^t}(s_h, a_h)] \leq \frac{C' r_{\max}^2 \log(TH|\mathcal{F}|/\delta)}{K}$$

*where $C'$ is an absolute constant.*

Now, we bound term (2). Note that

$$\mathbb{E}_{s_h \sim d_h^{\pi^*}}[\langle Q_h^{\pi^t}(s_h) - \hat{Q}_h^{\pi^t}(s_h), \pi_h^*(s_h) - \pi_h^t(s_h) \rangle]$$

$$\leq |\mathbb{E}_{s_h \sim d_h^{\pi^*}}[Q_h^{\pi^t}(s_h) - \hat{Q}_h^{\pi^t}(s_h), \pi_h^*(s_h)]| + |\mathbb{E}_{s_h \sim d_h^{\pi^*}}[Q_h^{\pi^t}(s_h) - \hat{Q}_h^{\pi^t}(s_h), \pi_h^t(s_h)]|$$

$$= |\mathbb{E}_{(s_h, a_h) \sim d_h^{\pi^*}}[Q_h^{\pi^t}(s_h, a_h) - \hat{Q}_h^{\pi^t}(s_h, a_h)]|$$

$$\quad + |\mathbb{E}_{s_h \sim d_h^{\pi^*}, a_h \sim \pi_h^t(a_h|s_h)}[Q_h^{\pi^t}(s_h, a_h) - \hat{Q}_h^{\pi^t}(s_h, a_h)]|$$

$$\leq \sqrt{\mathbb{E}_{(s_h, a_h) \sim d_h^{\pi^*}}\left[\left(Q_h^{\pi^t}(s_h, a_h) - \hat{Q}_h^{\pi^t}(s_h, a_h)\right)^2\right]}$$

$$\quad + \sqrt{\mathbb{E}_{s_h \sim d_h^{\pi^*}, a_h \sim \pi_h^t(a_h|s_h)}\left[\left(Q_h^{\pi^t}(s_h, a_h) - \hat{Q}_h^{\pi^t}(s_h, a_h)\right)^2\right]}$$

$$\leq 2\sqrt{\left(\max_{h \in [H]} \sup_{(s,a) \in \mathcal{S} \times \mathcal{A}} w(s, a, h)\right) \mathbb{E}_{(s_h, a_h) \sim d_h^\rho}\left[\left(Q_h^{\pi^t}(s_h, a_h) - \hat{Q}_h^{\pi^t}(s_h, a_h)\right)^2\right]}$$

$$\leq \frac{C r_{\max}^2 \log(TH|\mathcal{F}|/\delta)}{K} \sqrt{\max_{h \in [H]} \sup_{(s,a) \in \mathcal{S} \times \mathcal{A}} w(s, a, h, \gamma)}$$

where $w(s, a, h, \gamma) = \frac{d_h^{\pi^*}(s,a)}{d_h^\rho(s,a)}$ is the density ratio between $d_h^{\pi^*}(s, a)$ and $d_h^\rho(s, a)$ and $C$ is a constant.

Therefore, with probability $1 - \delta$, we have

$$\sum_{t=1}^{T} \sum_{h=0}^{H-1} \mathbb{E}_{s_h \sim d_h^{\pi^*}} [\langle Q_h^{\pi^t}(s_h) - \hat{Q}_h^{\pi^t}(s_h), \pi_h^*(s_h) - \pi_h^t(s_h)\rangle] \tag{11}$$

$$\leq TH \frac{Cr_{\max}^2 \log(TH|\mathcal{F}|/\delta)}{K} \sqrt{\max_{h \in [H]} \sup_{(s,a) \in \mathcal{S} \times \mathcal{A}} w(s,a,h,\gamma)} \tag{12}$$

$$\tag{13}$$

Now, we would like to show that the density ratio will decrease with an increase in $\gamma$.

Based on Theorem 1 in [70], we can rewrite $d_h^\rho(s,a)$ as $d_h^\rho(s,a) = d_h^{\pi^t}(s,a)\pi^{t+1}(a|s)^\gamma$ The density ratio $w(s,a)$ is

$$\frac{d_h^{\pi^*}(s,a) \sum_{\mathcal{A}} d_h^{\pi^t}(s,a)\pi^{t+1}(a|s)^\gamma}{d_h^{\pi^t}(s,a)\pi^{t+1}(a|s)^\gamma}$$

Suppose $d_h^{\pi^t}(s,a) \propto \exp(\beta_1 A^{\pi^*}(s,a))$ and $\pi^{t+1}(a|s) \propto \exp(\beta_2 A^{\pi^*}(s,a))$ for some parameter $\beta_1 < \beta_2$. This is reasonable since the updated policy $\pi^{t+1}$ is better than the current policy $\pi^t$. Take the logarithm and we have

$$\log w(s,a,h,\gamma) = \log d_h^{\pi^*}(s,a) + \log\left(\sum_{\mathcal{A}} e^{(\beta_1 + \beta_2\gamma)A^{\pi^*}(s,a)}\right) - (\beta_1 + \beta_2\gamma)A^{\pi^*}(s,a) \tag{14}$$

Partial derivative of $\log w(s,a,h,\gamma)$ with respect to $\gamma$ is:

$$\frac{\partial}{\partial\gamma}(\log w(s,a,h,\gamma)) = \beta_2 \left[\frac{\sum_{\mathcal{A}} A^{\pi^*}(s,a)e^{(\beta_1 + \beta_2\gamma)A^{\pi^*}(s,a)}}{\sum_{\mathcal{A}} e^{(\beta_1 + \beta_2\gamma)A^{\pi^*}(s,a)}} - A^{\pi^*}(s,a)\right] \tag{15}$$

Note that the largest density ratio happens for $a^* = argmax_a A^{\pi^*}(s,a)$. Due to the softmax function in the gradient, we see that for $a^*$, the derivative is negative, meaning that by increasing $\gamma$, the regret bound will decrease.

## C.2 Preference optimization

*Proof.* We follow the proof strategy outlined in Theorem 6.1 of [21]. To derive the desired result, we begin with the key observation that DPO [9] is equivalent to optimizing a KL-regularized expected reward objective, where the reward function is used to define preferences via the Bradley-Terry model. Specifically, the optimal policy $\pi^*(\cdot \mid \cdot)$ that maximizes the following regularized objective:

$$\max_\pi \mathbb{E}_{x \sim \mu, y \sim \pi(\cdot|x)} [r(x,y)] - \beta D_{\text{KL}}(\pi(\cdot \mid x) \| \pi_{\text{ref}}(\cdot \mid x))$$

is given in closed form as:

$$\pi^*(y \mid x) \propto \pi_{\text{ref}}(y \mid x) \cdot \exp\left(\frac{r(x,y)}{\beta}\right). \tag{16}$$

This optimal policy can be recovered by applying DPO to preference-labeled pairs $(x, y_1, y_2)$, where preferences are sampled from the Bradley-Terry model [49] defined by the reward function $r$:

$$p(y_1 \succeq y_2 \mid x) = \frac{\exp(r(x,y_1))}{\exp(r(x,y_1)) + \exp(r(x,y_2))}. \tag{17}$$

Given this background, we consider preference pairs of the form $(x, [y_{0:i-1}, y_i^+], [y_{0:i-1}, y_i^-])$, where both continuations are sampled from the $\pi_{ref}$: $y_i^+ \sim \pi_{\text{ref}}(\cdot \mid x, y_{0:i-1}), \quad y_i^- \sim \pi_{\text{ref}}(\cdot \mid x, y_{0:i-1})$, and the preference is determined based on the advantage estimates $A^{\pi_{ref}}(x, y_{0:i-1}; \cdot)$, Combined with Eqn. 16, this yields the following equation:

$$\pi_i(y_i \mid x, y_{0:i-1}) \propto \pi_{\text{ref}}(y_i \mid x, y_{0:i-1}) \cdot \exp\left(\frac{A^{\pi_{ref}}(x, y_{0:i-1}; y_i)}{\beta}\right). \tag{18}$$

Moreover, since the optimal advantage-weighted RL policy that maximizes Eqn. 18 coincides with the solution in Eqn. 4, the proof is complete.

## D  Additional Preference Optimization Objectives

Beyond the DPO objective described in the main text, several other preference optimization methods have been developed. These methods also typically operate on an offline dataset of preference pairs $\mathcal{D} = \{(x, y^+, y^-)\}$, where $y^+$ is the preferred response and $y^-$ is the dispreferred response given an input $x$. The policy being optimized is denoted by $\pi_\theta$, with $\theta$ being its parameters, and $\pi_{\text{ref}}$ is a fixed reference policy.

ORPO [71] introduces an objective that penalizes the model for assigning low likelihood to preferred responses, while simultaneously ensuring that preferred responses have higher odds than dispreferred ones. For the single preference pair $(x, y^+, y^-)$, the ORPO loss is:

$$\mathcal{L}_{\text{ORPO}}(\pi_\theta; x, y^+, y^-) = -\log p_\theta(y^+|x) - \lambda \log \sigma \left( \log \frac{p_\theta(y^+|x)}{1 - p_\theta(y^+|x)} - \log \frac{p_\theta(y^-|x)}{1 - p_\theta(y^-|x)} \right),$$

where $p_\theta(y|x) = \exp\left(\frac{1}{|y|} \log \pi_\theta(y|x)\right)$ is a length-normalized likelihood for sequence $y$, $\log \pi_\theta(y|x)$ is the sum of log-probabilities of tokens in $y$, $\sigma(\cdot)$ is the sigmoid function, and $\lambda$ is a weighting coefficient. This formulation directly encourages the policy to generate $y^+$ and ensures its odds are favorable compared to $y^-$.

SimPO [47] offers a modification of the DPO-style loss by incorporating sequence length normalization directly into the log-probability difference and adding a margin term $\gamma$. The SimPO loss for a preference pair is:

$$\mathcal{L}_{\text{SimPO}}(\pi_\theta; x, y^+, y^-) = -\log \sigma \left( \beta \frac{\log \pi_\theta(y^+|x)}{|y^+|} - \beta \frac{\log \pi_\theta(y^-|x)}{|y^-|} - \gamma \right),$$

where $\beta$ is a constant that controls the scaling of the reward difference. This objective aims to maximize the margin between the length-normalized log-likelihood of the preferred response and that of the dispreferred response.

KTO [46] introduces an alignment approach rooted in prospect theory. Instead of optimizing preference likelihoods, KTO focuses on directly maximizing the utility of each individual generation $y$ given an input $x$. A distinctive characteristic is its reliance on a binary signal for every input-output pair $(x, y) \in \mathcal{D}$, classifying $y$ as either desirable or undesirable for $x$. Consequently, KTO does not inherently require paired preference data (*i.e.,* $y^+$ vs $y^-$). The loss for a single such sample $(x, y)$ is formulated to be minimized and is given by:

$$\mathcal{L}_{\text{KTO}}(\pi_\theta, \pi_{\text{ref}}; x, y) = \begin{cases} \lambda_D \left(1 - \sigma \left(\beta \left(r_\theta(x, y) - z_0\right)\right)\right) & \text{if } y \text{ is desirable for } x \\ \lambda_U \left(1 - \sigma \left(\beta \left(z_0 - r_\theta(x, y)\right)\right)\right) & \text{if } y \text{ is undesirable for } x \end{cases}$$

In this formulation, $r_\theta(x, y) = \log \frac{\pi_\theta(y|x)}{\pi_{\text{ref}}(y|x)}$ represents the log-probability ratio of the current policy $\pi_\theta$ against a fixed reference policy $\pi_{\text{ref}}$. The term $z_0$ serves as a reference point related to the KL estimate. $\beta$ is a hyperparameter modulating risk aversion, and $\lambda_D, \lambda_U$ are positive hyperparameters that weight the contributions from desirable and undesirable outputs, respectively. This structure allows KTO to process feedback on individual generations, and if paired data is available, each part of the pair $(y^+, y^-)$ would contribute its own loss term based on its desirable/undesirable status.

## E  Additional Experiment Details

In this section, we provide additional details on the experiments.

### E.1  Dataset Details

As mentioned in §6, we utilize established train/test splits for several benchmarks. For GSM8K, MATH, MMLU, and MMLUPro, we adopt their standard train/test distributions. Specifically for BBH, the dataset is randomly partitioned into training and test sets at the sub-task level. For the AIME dataset, problems from the years 1983-2023 constitute the training set, while problems from 2024-2025 form the test set. Detailed statistics for each dataset, including the number of samples in the training and test sets, and the source of the dataset, are presented in Table 2.

Table 2: **Dataset Statistics**: The table presents the number of training and test samples for each dataset, along with the source of the dataset.

| Dataset | # Train Samples | # Test Samples | Source |
|---|---|---|---|
| GSM8K | 7470 | 1320 | https://huggingface.co/datasets/openai/gsm8k |
| MATH | 12000 | 500 | https://github.com/openai/prm800k |
| MMLU | 99842 | 14042 | https://huggingface.co/datasets/cais/mmlu |
| MMLUPro | 9625 | 2407 | https://huggingface.co/datasets/TIGER-Lab/MMLU-Pro |
| BBH | 3261 | 3250 | https://github.com/suzgunmirac/BIG-Bench-Hard |
| AIME-2024 | 903 | 30 | https://huggingface.co/datasets/gneubig/aime-1983-2024 |
| AIME-2025 | 903 | 30 | https://huggingface.co/datasets/yentinglin/aime_2025 |

## E.2 Implementation Details

Here we provide further details on the data processing and algorithm implementation for the experiments.

**Question Filtering**    To construct a training set of appropriate difficulty, we apply a filtering process to each dataset. For every question in the initial training pool, we generate 8 responses using the base model with a sampling temperature of 0.7. Questions that are solved correctly across all eight attempts, or conversely, incorrectly across all eight attempts, are subsequently excluded from the training set. This procedure aims to retain questions that are neither trivially easy nor prohibitively difficult for the base model, thereby focusing the fine-tuning process on a more informative problem distribution. This filtering protocol is applied uniformly to create the training data for both the baseline methods and their GPO-enhanced counterparts.

**Step Grouping for Trajectory Segmentation**    Reasoning trajectories generated by the models are segmented into multi-steps using the following procedure. First, each trajectory is split into steps based on newlines. Multiple consecutive newline characters are collapsed into a single newline. To avoid overly short steps, any step consisting of fewer than 30 words is merged with the previous step. Furthermore, to maintain computational feasibility, we impose a maximum number of steps: 15 for offline preference optimization methods and 10 for online PPO. If a trajectory exceeds this maximum step count, all subsequent steps beyond the limit are concatenated to form the last step.

**Preference Data Preparation for Offline Methods**    For the offline preference optimization algorithms DPO, SimPO, and ORPO, positive and negative preference data are sourced during the question filtering process described above. Since the filtering retains questions for which the base model produces a mix of correct and incorrect responses across the 8 generated samples, these naturally provide pairs of successful (positive) and unsuccessful (negative) trajectories for the same input question.

For KTO, which requires demonstration data rather than explicit preference pairs, we adapt the approach from the original KTO paper [46]. For the baseline KTO, each preference pair $(y^+, y^-)$ from the DPO dataset is decomposed into two separate demonstrations: $y^+$ with the positive tag and $y^-$ with the negative tag. For the GPO-enhanced KTO, we first apply the GPO strategy to form paired preference data. These $n$ preference pairs are then similarly decomposed into $2 * n$ individual positive and negative demonstrations to train the GPO-enhanced KTO model.

## E.3 Hyper-parameters for Experiments

We list the main hyper-parameters for the experiments in Table 3. The GPO-enhanced methods are trained with the same hyper-parameters as the baseline methods. For those unmentioned hyper-parameters, we use the default values provided by the LLaMA-Factory framework [72].

## E.4 Running Environment

Our training process primarily utilizes the LLaMA-Factory framework [72]. The experiments are conducted on a server equipped with four AMD EPYC 7702 64-Core CPU Processors and eight NVIDIA H100 80GB GPUs. The total computational resources consumed include approximately 1800 GPU hours and 2 TB of storage for model checkpoints.

Table 3: **Hyper-parameters for Different Datasets and Methods.**

| Method | Hyper-parameter | Dataset | | | | | | |
|---|---|---|---|---|---|---|---|---|
| | | GSM8K | MATH | MMLU | MMLUPro | BBH | AIME-2024 | AIME-2025 |
| **Common** | LoRA Alpha ($\alpha$) | 2 | 2 | 2 | 2 | 2 | 2 | 2 |
| | LoRA Rank | 8 | 8 | 8 | 8 | 8 | 8 | 8 |
| | LoRA Target | all | all | all | all | all | all | all |
| | Optimizer | AdamW | AdamW | AdamW | AdamW | AdamW | AdamW | AdamW |
| | Sequence Length | 4096 | 4096 | 2048 | 2048 | 2048 | 4096 | 4096 |
| **PPO** | Clip Value | 0.2 | 0.2 | 0.2 | 0.2 | 0.2 | 0.2 | 0.2 |
| | KL Divergence Coeff. | 0.05 | 0.05 | 0.05 | 0.05 | 0.05 | 0.05 | 0.05 |
| | Learning Rate | 1e-5 | 1e-5 | 2e-5 | 2e-5 | 1e-5 | 1e-5 | 1e-5 |
| | Batch Size | 32 | 64 | 64 | 64 | 128 | 32 | 32 |
| | Epochs | 5 | 5 | 2 | 3 | 5 | 5 | 5 |
| **DPO** | Beta ($\beta$) | 0.1 | 0.1 | 0.1 | 0.1 | 0.1 | 0.1 | 0.1 |
| | Learning Rate | 1e-5 | 1e-5 | 1e-5 | 1e-5 | 1e-5 | 1e-5 | 1e-5 |
| | Batch Size | 4 | 16 | 16 | 16 | 8 | 4 | 4 |
| | Epochs | 10 | 10 | 5 | 10 | 10 | 10 | 10 |
| **KTO** | Desirable Reward Scalar | 1.0 | 1.0 | 1.0 | 1.0 | 1.0 | 1.0 | 1.0 |
| | Undesirable Reward Scalar | 1.0 | 1.0 | 1.0 | 1.0 | 1.0 | 1.0 | 1.0 |
| | Learning Rate | 1e-5 | 1e-5 | 1e-5 | 1e-5 | 1e-5 | 1e-5 | 1e-5 |
| | Batch Size | 8 | 16 | 16 | 16 | 8 | 8 | 8 |
| | Epochs | 10 | 10 | 2 | 10 | 10 | 10 | 10 |
| **SimPO** | Reward Margin | 0.5 | 0.5 | 0.5 | 0.5 | 0.5 | 0.5 | 0.5 |
| | Learning Rate | 1e-5 | 5e-6 | 5e-6 | 1e-5 | 1e-5 | 1e-5 | 1e-5 |
| | Batch Size | 4 | 16 | 16 | 16 | 8 | 4 | 4 |
| | Epochs | 10 | 10 | 5 | 10 | 10 | 10 | 10 |
| **ORPO** | Learning Rate | 1e-5 | 1e-5 | 1e-5 | 1e-5 | 1e-5 | 1e-5 | 1e-5 |
| | Batch Size | 2 | 16 | 16 | 16 | 8 | 2 | 2 |
| | Epochs | 10 | 10 | 5 | 10 | 10 | 10 | 10 |

Table 4: Accuracy on a long-context reasoning subset of BBEH.

| Method | Accuracy (%) |
|---|---|
| Base Model (DeepSeek-R1-Distill-Qwen-7B) | 24.5 |
| DPO Baseline | 29.0 |
| **GPO-DPO (with step grouping)** | **36.0** |

### E.5 Experimental Details for Long-Trajectory Reasoning

We provide supplementary details for the experiment discussed in Section 8, which was designed to validate the scalability of GPO to tasks involving very long trajectories.

**Experimental Setup.** To create a challenging long-context reasoning benchmark, we selected a sub-dataset from BIG-Bench Extra Hard (BBEH). This dataset was chosen for its particularly long problem descriptions (averaging approximately 1,700 tokens) and lengthy, complex reasoning chains (averaging approximately 190 lines). The experiment was conducted using a training set of 1,000 questions and a test set of 200 questions. The base model used for all experiments was DeepSeek-R1-Distill-Qwen-7B.

**Step Grouping Heuristic.** To adapt GPO for these long trajectories without incurring prohibitive computational costs, we implemented a simple step grouping heuristic. For the GPO-DPO method, we grouped every 15 lines of the generated reasoning chain into a single logical "step". This approach maintains the core principle of identifying pivotal moments for targeted feedback while significantly reducing the number of points at which future returns need to be estimated.

**Results.** The results, summarized in Table 4, demonstrate that even with this straightforward heuristic, GPO delivers substantial performance gains over both the base model and the DPO baseline. This confirms that GPO can be effectively adapted for long-trajectory reasoning tasks.

# F    User Study

To evaluate if the human annotators agree with the critical steps identified by our method, we conducted a user study. This section details the setup of the study and the format used for presenting questions to the participants.

## F.1    User Study Setup

Before participating in the study, participants are presented with an informed consent form, outlining the purpose of the research, the nature of their participation, and confirming that no private information would be collected. Participation is voluntary.

The study consisted of five questions and is designed to take approximately five minutes to complete. For each question, participants were presented with the following:

- **Task description:** A brief overview of the problem type.
- **Question:** The specific question posed to the LLM.
- **Ground truth answer:** The correct answer to the question.
- **LLM's reasoning trajectory:** The step-by-step reasoning process generated by an LLM, with each step clearly numbered. This reasoning leads to an incorrect answer.
- **Selection task:** Participants are then shown four potential critical steps. Their task is to select the single step they believe is the most critical in LLM's reasoning towards the result. Within the four steps, there is one step that is identified by our method, and the other three steps are randomly selected from the rest of the steps in the trajectory.

## F.2    User Study Questions and Results Format

We list the questions and the results format below. We highlight the critical step identified by our method in red for the paper presentation, and note that the participants will not see the red-highlighted text.

---

**Question 1: Infer the date from context**

**Task description:** Infer the date from context.

**Q:** It was Sept. 1st, 2021 a week ago. What is the date 24 hours later in MM/DD/YYYY?

**Options:**
(A) 03/09/2022
(B) 09/10/2021
(C) 08/31/2021
(D) 09/09/2021 (Ground Truth)
(E) 09/02/2021
(F) 09/17/2021

**The Large language model's reasoning trajectory with labeled step number is:**
1. Okay, so I need to figure out the date 24 hours after September 1st, 2021. Let me break this down step by step.
2. First, the question says it was September 1st, 2021, a week ago. Wait, does that mean today is September 8th, 2021? Because a week is seven days, so adding seven days to September 1st would land on September 8th. But the question is asking about 24 hours after September 1st, not today's date. Hmm, maybe I'm overcomplicating it.
3. Wait, no, the question is straightforward. It says, 'It was Sept. 1st, 2021 a week ago. What is the date 24 hours later in MM/DD/YYYY?" So, regardless of today's date, I just need to find the date that is exactly 24 hours after September 1st, 2021.
4. So, 24 hours later would be the next day, September 1st, plus one day is September 2nd, 2021. Let me check the options provided.
5. (A) 03/09/2022-That's March 9th, 2022. That's way too far in the future.

---

6. (B) 09/10/2021-September 10th, 2021. That's nine days later, not just one.

7. (C) 08/31/2021-August 31st, 2021. That's the day before September 1st, so that's a day earlier, not later.

8. (D) 09/09/2021-September 9th, 2021. Again, that's eight days later.

9. (E) 09/02/2021-September 2nd, 2021. That's exactly one day after September 1st.

10. (F) 09/17/2021-September 17th, 2021. That's 16 days later.

11. So, the correct answer should be option (E), which is 09/02/2021. I think I got it right. Just adding one day to September 1st gives September 2nd.

12. The date 24 hours after September 1st, 2021 is September 2nd, 2021, which corresponds to option (E).

**Participant task:** There are several steps listed; please select which step you prefer that is more critical to the LLM's failure.
- Step 2
- Step 3
- Step 5
- Step 11

**Results for Question 1:**
- Step 2(Our method): 44%
- Step 3: 32%
- Step 5: 6%
- Step 11: 18%

## Question 2: Answer questions about causal attribution

**Task description:** Answer questions about causal attribution.

**Q:** How would a typical person answer each of the following questions about causation? Drew, Kylie, Oliver, and Jen are regular customers at a small, local coffee shop. Given the selling price of the coffee and the cost of daily operation, the coffee shop will turn a profit if anyone orders coffee on a given day. Only one person ordering coffee is needed for the coffee shop to turn a profit that day. Drew, Kylie, Oliver, and Jen all usually order coffee on Tuesdays. This Tuesday, as usual, Drew ordered coffee. The same day, Kylie ordered coffee, Oliver ordered coffee, and Jen also ordered coffee. Since at least one person ordered coffee on Tuesday, the coffee shop made a profit that day. Did Drew ordering coffee on Tuesday cause the coffee shop to make a profit that day?

**Options:**
- Yes (Ground Truth)
- No

**The Large language model's reasoning trajectory with labeled step number is:**

1. Okay, so I'm trying to figure out whether Drew ordering coffee on Tuesday caused the coffee shop to make a profit that day. Let me break this down step by step.

2. First, the coffee shop makes a profit if at least one person orders coffee. So, the key here is that only one order is needed for a profit. Now, on this particular Tuesday, Drew, Kylie, Oliver, and Jen all ordered coffee. That means there were four orders that day.

3. The question is specifically about Drew's order. Did Drew's action cause the profit? Well, since the shop only needs one order to make a profit, Drew's order alone would have been sufficient. But in this case, there were three other people who also ordered coffee. So, even if Drew hadn't ordered, the shop would still have made a profit because of the others.

4. This makes me think about the concept of causation. For Drew's order to be a cause, it should be necessary for the outcome. But since the outcome (profit) would have happened regardless of Drew's order, his action isn't the necessary cause. It's more like a contributing factor, but not the sole or necessary one.

5. In this case, Drew's order is sufficient on its own, but it's not necessary because others also ordered. So, while his order contributed, it wasn't the cause in the sense that it was required for the profit.
6. Another angle is to think about what would happen if Drew didn't order. If the shop still made a profit because of the others, then Drew's order wasn't the cause. It's like saying if you have multiple people contributing to an event, each one isn't the sole cause unless their absence would change the outcome.
7. So, putting it all together, Drew's order wasn't the cause because the profit would have occurred without him. Therefore, the answer should be no.
8. The coffee shop made a profit because at least one person ordered coffee, but since others also ordered, Drew's order wasn't necessary for the profit. Therefore, Drew's action wasn't the cause.

**Participant task:** There are several steps listed; please select which step you prefer that is more critical to the LLM's failure.
- Step 1
- Step 4
- Step 5
- Step 6

**Results for Question 2:**
- Step 1: 2%
- Step 4(Our method): 68%
- Step 5: 22%
- Step 6: 8%

---

**Question 3: Questions that involve enumerating objects and asking the model to count them**

**Task description:** Questions that involve enumerating objects and asking the model to count them.

**Q:** I have a microwave, a lamp, four cars, a stove, a toaster, and a bed. How many objects do I have?

**Ground Truth Answer:** 9

**The Large language model's reasoning trajectory with labeled step number is:**
1. Okay, so I need to figure out how many objects the person has. Let me list them out one by one to make sure I don't miss anything.
2. First, they mention a microwave. That's one object. Then there's a lamp, so that's two.
3. Next, they have four cars. Wait, four cars would count as four separate objects, right? So that adds four to the total.
4. After that, there's a stove, which is another object, making it five so far. Then a toaster, that's six. Finally, a bed, which brings the total to seven.
5. Let me double-check to make sure I didn't skip any. Microwave (1), lamp (2), four cars (4), stove (5), toaster (6), bed (7). Yep, that adds up to seven objects in total.
6. The person has a microwave, a lamp, four cars, a stove, a toaster, and a bed. Counting each item individually, the total number of objects is 7.

**Participant task:** There are several steps listed; please select which step you prefer that is more critical to the LLM's failure.
- Step 2
- Step 4
- Step 5
- Step 6

**Results for Question 3:**

## Question 4: Multi-step arithmetic word problem

**Task description:** Multi-step arithmetic word problem.

**Q:** Jeremy is driving 600 miles to visit his parents. His average speed for this trip is 50 miles per hour. Every two hours of driving, he makes a rest stop for 15 minutes. His car gets 18 miles per gallon of gas, and his gas indicator comes on when he's used 15 gallons, at which point he stops to refill, which takes 10 minutes. How long, in minutes, does he take to get to his parents' house if he starts out with a full tank of gas?

**Ground Truth Answer:** 815

**The Large language model's reasoning trajectory with labeled step number is:**
1. First, I need to determine how long it will take Jeremy to drive 600 miles at an average speed of 50 miles per hour. Next, I'll calculate the total driving time by dividing the total distance by his average speed.
2. Then, I'll figure out how many rest stops he'll need. Since he takes a 15-minute break every two hours of driving, I'll divide the total driving time by 2 to find the number of rest stops. After that, I'll calculate the total rest stop time by multiplying the number of rest stops by 15 minutes.
3. Next, I'll determine how much gas he uses during the trip by dividing the total distance by his car's fuel efficiency. Then, I'll find out how many times he needs to refill his gas tank by dividing the total gallons used by the capacity of his gas tank.
4. Each refill takes 10 minutes, so I'll multiply the number of refills by 10 to get the total gas stop time. Finally, I'll add up the driving time, rest stop time, and gas stop time to get the total time for the trip in minutes.
5. Let's break down Jeremy's trip step by step to determine the total time he takes to reach his parents' house. ###1. Calculate the Total Driving Time Distance to Travel: 600 miles Average Speed: 50 miles per hour Driving Time = Distance/Speed = 600 miles/50 mph = 12 hours
6. ###2. Determine the Number of Rest Stops Jeremy makes a rest stop every 2 hours of driving. Number of Rest Stops = 12 hours/2 hours per stop = 6 stops. Each rest stop lasts 15 minutes. Total Rest Stop Time = 6 stops * 15 minutes = 90 minutes = 1.5 hours
7. ###3. Calculate the Gas Consumption and Stops Car Fuel Efficiency: 18 miles per gallon Total Gas Used: Total Gas = 600 miles/18 mpg = 33.33 gallons
8. Number of Gas Stops: Since the gas indicator comes on after using 15 gallons, we divide the total gas by the gas used per stop. Number of Gas Stops = 33.33 gallons/15 gallons = 2.22 stops
9. However, since he starts with a full tank, he will need to refill before the gas runs out. Therefore, he makes 2 gas stops. Each gas stop takes 10 minutes. Total Gas Stop Time = 2 stops * 10 minutes = 20 minutes = 1/3 hour
10. ###4. Calculate the Total Time Total Time = Driving Time + Rest Stop Time + Gas Stop Time Total Time = 12 hours + 1.5 hours + 1/3 hours = 13.833 hours Convert the total time to minutes: 13.833 hours * 60 minutes/hour = 830 minutes

**Participant task:** There are several steps listed; please select which step you prefer that is more critical to the LLM's failure.
- Step 3
- Step 4
- Step 6
- Step 9

**Results for Question 4:**
- Step 3: 14%
- Step 4: 10%
- Step 6(Our method): 76%
- Step 9: 0%

---

## Question 5: Financial calculation word problem

**Task description:** Financial calculation word problem.

**Q:** Tom's cat needs an expensive surgery. He has had pet insurance for 24 months that costs $20 per month. The procedure cost $5000, but the insurance covers all but 20% of this. How much money did he save by having insurance?

**Answer:** 3520 (Ground Truth)

**The Large language model's reasoning trajectory with labeled step number is:**
1. First, I need to calculate the total cost of Tom's pet insurance over 24 months. The insurance costs $20 per month, so multiplying that by 24 gives the total insurance expenditure.
2. Next, I'll determine the cost of the surgery. The surgery itself costs $5000, but Tom's pet insurance covers all but 20% of this amount. To find the amount covered by insurance, I'll calculate 20% of $5000 and subtract that from the total surgery cost to get the out-of-pocket expense.
3. After that, I'll compare the total insurance cost with the out-of-pocket surgery cost to find the savings. By subtracting the out-of-pocket expense from the insurance cost, I can determine how much money Tom saved by having insurance. Finally, I'll present the calculated savings as the answer.
4. Let's break down the problem step by step to determine how much money Tom saved by having pet insurance. ###Step 1: Calculate the Total Pet Insurance Cost Tom has pet insurance for 24 months at a cost of $20 per month.
5. Total Insurance Cost = 24 months * $20/month = $480 ###Step 2: Determine the Surgery Cost Covered by Insurance The surgery costs $5000, and the insurance covers all but 20% of this amount.
6. Insurance Coverage = 100% - 20% = 80% Amount Covered by Insurance = 80% * $5000 = 0.8 * $5000 = $4000
7. ###Step 3: Calculate the Out-of-Pocket Expense Tom will pay the remaining 20% of the surgery cost out of pocket. Out-of-Pocket Expense = 20% * $5000 = 0.2 * $5000 = $1000
8. ###Step 4: Determine the Savings To find out how much Tom saved by having insurance, subtract the out-of-pocket expense from the total insurance cost. Savings = Total Insurance Cost - Out-of-Pocket Expense = $480 - $1000 = -$520
9. Wait, this result indicates that Tom actually spent $520 more with insurance compared to not having insurance. However, this doesn't make sense because insurance should reduce his expenses. Let's re-examine the calculations. Revised Approach The correct way to calculate savings is:
10. The correct way to calculate savings is: Savings = Surgery Cost - (Surgery Cost - Insurance Coverage) - Total Insurance Cost Savings = $5000 - ($5000 * 0.8) - $480 = $5000 - $4000 - $480 = $520

**Participant task:** There are several steps listed; please select which step you prefer that is more critical to the LLM's failure.
- Step 4
- Step 6
- Step 8
- Step 10

**Results for Question 5:**

- Step 4: 6%
- Step 6: 8%
- Step 8(Our method): 56%
- Step 10: 30%

## F.3  Overall Findings

Overall, the results indicate a strong alignment between the critical steps identified by GPO and human judgment. Across the five questions evaluated, the percentage of participants who selected the GPO-identified step as the most critical was 44%, 68%, 88%, 76%, and 56%, respectively. The strong alignment between the steps identified by our method and those recognized by humans as critical points indicates that our process effectively highlights key reasoning steps. This alignment serves as qualitative validation for the core mechanism of GPO, reinforcing the hypothesis that its empirical improvements are well-founded.

