# OpenReview forum: "GPO: Learning from Critical Steps to Improve LLM Reasoning"
_NeurIPS.cc/2025/Conference — NeurIPS 2025 poster_

### Official Review · Reviewer_r5W8 · 2025-06-28

**Clarity:** 3
**Significance:** 2
**Originality:** 2
**Rating:** 4
**Confidence:** 4

**Summary:**

This paper introduces Guided Pivotal Optimization (GPO). GPO stands as a general strategy integrable to existing optimization methods by identifying critical steps and prioritizing the learning process on the rollouts of these critical steps, which are identified by estimating the advantage functions per step. This work also provides a theoretical analysis that proves the online GPO as a form of advantage-weighted RL in the offline preference learning setting. The evaluation is extensive, where GPO demonstrates consistent improvements over optimization baselines. Lastly, it includes a user study with 50 participants analyzing the alignment between the GPO critical step identification and human judgment.

**Questions:**

1. Could you also include the standard deviation in Table 1?

2. How would you justify splitting critical steps by newlines?

3. I see you set a maximum number of steps around 10 to 15. However, for logical reasoning methods, can you justify whether your method scales to tasks with a larger number of steps?

**Ethical Concerns:**

["NO or VERY MINOR ethics concerns only"]

**Final Justification:**

The rebuttal has addressed my questions for this work. Given the limited backbone models studied in this work and the heuristics of the newline separation for steps, I am keeping my rating at 4 with an improved confidence.

**Limitations:**

Yes.

**Paper Formatting Concerns:**

No.

**Quality:**

3

**Strengths And Weaknesses:**

Strength:

1. Automatically identifying the critical step via advantage in an online learning setup is integrable with many existing optimization methods.

2. The theoretical analysis is useful in bridging the gap between the proposed online learning and the offline preference learning.

3. The evaluation dataset and baselines provide a good coverage of tasks and existing learning approaches to reflect the generalization of the proposed method.

Weakness:

1. As acknowledged by the author, integrating GPO inevitably introduces roughly twice the computation cost.

2. The critical steps are classified by splitting the new line in Algorithm 1. I find this definition may be adequate for mathematical reasoning tasks. However, I doubt the generalization of this definition to a wider array of tasks, such as the coding task or factual QA tasks.

3. The evaluation is only conducted on a single base model DeepSeek-R1-Distill-Qwen-7B model. It is still necessary to conduct the experiments on other families of models to provide sufficient evidence of the generalization of GPO.

---

> ### Author Rebuttal · Authors · 2025-07-30
>
> We thank Reviewer r5W8 for the constructive and insightful comments. Please see our response to each of your questions below.
>
> > I doubt the generalization of this definition to a wider array of tasks, such as the coding task or factual QA tasks. How would you justify splitting critical steps by newlines.
>
> **Response:** This is a good point, and we thank the reviewer for raising it. Our choice to split steps by newlines is a practical heuristic for reasoning tasks where LLMs naturally structure their thoughts into distinct lines. In LLM generation, there is no inherent "step" as in traditional reinforcement learning. While generation is token-by-token, treating each token as a step is computationally expensive and semantically weak, as individual tokens rarely carry complete meaning. Therefore, we use newlines to segment the output into semantical steps of reasoning, and this strategy can also be found at prior works[1,2]
>
> We agree that this specific heuristic may not be optimal for all tasks. However, the core idea of GPO is to identify the critical step and explore, where the definition of a ‘step’ can be flexible and application-dependent. For example, for the coding task, a step can be the generation of a complete function, a class or a code block. For long-form QA, a step could be a full paragraph.
>
> The step-grouping heuristic we discuss below is a prime example of this adaptability, demonstrating that the step can be adjusted to suit the task’s complexity and structure. We will make this point about the flexibility of step definition explicit in the revised version.
>
> [1] Wang, Huaijie, et al. "Offline reinforcement learning for llm multi-step reasoning." Association for Computational Linguistics: ACL 2025
>
> [2] Setlur, Amrith, et al. "Rewarding progress: Scaling automated process verifiers for llm reasoning." ICLR 2025
>
> > Can you justify whether your method scales to tasks with a larger number of steps?
>
> **Response:** To address the challenge of very long trajectories, we employ an initial and simple heuristic: grouping multiple generation lines into a single step. This maintains the core idea of identifying pivotal moments without incurring significant computational cost for complex trajectories.
>
> To validate this, we conducted a new experiment on a challenging reasoning task. We used a sub-dataset 'boardgame' from BIG-Bench Extra Hard (BBEH) as it has the longest problem descriptions within BBEH (avg. ~1700 tokens) and lengthy reasoning chains to solve the problem (avg. ~190 lines). We randomly selected another 5 subtasks as the training set. The test set contains 200 questions while the train set contains 1000 questions. We applied a simple heuristic of grouping every 15 lines into a single step for GPO. The results are as follows:
>
> -   **Base Model (DeepSeek-R1-Distill-Qwen-7B):** 28.5% accuracy
>
> -   **DPO Baseline:** 32.0% accuracy
>
> -   **GPO-DPO (with step grouping):**  **36.0% accuracy**
>
> This experiment confirms that GPO can be adapted for very long trajectories, yielding a substantial **+4% improvement** over DPO baseline. As grouping is a very initial mechanism, we would like to add the discussion and leave more complex strategies as future work.
>
> > Integrating GPO inevitably introduces roughly twice the computation cost.
>
> **Response:** Thanks for this comment. We think this can be alleviated by:
>
> 1.  **Managing Overhead via Simulation Samples:**
>
> As we discussed in Section 6.3 and demonstrated in Figure 3 (left), the computational overhead is manageable. Our experiments show that performance gains begin to saturate after a certain number of MC simulations. This allows for a practical balance between accuracy and computational cost, as one does not need an excessive number of rollouts to achieve most of the benefit.
>
> 2.  **Adapting to Long Trajectories with Step Grouping:** as we have shown.
>
> 3.  **Future Avenues for Efficiency:** Furthermore, there are promising avenues for future optimization. For the online setting (PPO-based), the current Monte Carlo estimation of the advantage function could be replaced with a more sample-efficient alternative, i.e., Generalized Advantage Estimation (GAE) [3], which may further reduce computational overhead. We can utilize the value network trained by the PPO algorithm to estimate the advantage more efficiently.
>
> We will incorporate the related discussion into the next version of our paper. We thank the reviewer again for their constructive feedback, which has helped us demonstrate the practical scalability of GPO.
>
> [3] Schulman, John, et al. "High-dimensional continuous control using generalized advantage estimation." arXiv preprint arXiv:1506.02438 (2015).
>
> > The evaluation is only conducted on a single base model.
>
> **Response:** We thank the reviewer for this valid point. To demonstrate that GPO’s effectiveness is not limited to a single model family, we have conducted new experiments on **Llama-3.1-8B** and **InternLM-3-8B** over two datasets: MATH and GSM8K. We selected three baselines for experiments. The results are shown below:
>
>  | Llama-3.1-8B  | GSM8K (%) | MATH (%) |
>  |--------|-----------|----------|
>  | Base | 69.60 | 62.80 |
>  | PPO | 72.36 | 67.20 |
>  | GPO-PPO | **74.17** | **69.80** |
>  | DPO | 76.57 | 68.20 |
>  | GPO-DPO | **77.30** | **72.40** |
>  | KTO | 75.39 | 68.60 |
>  | GPO-KTO | **76.57** | **74.20** |
>
>
> | InternLM-3-8B  | GSM8K (%) | MATH (%) |
> |--------|-----------|----------|
> | Base | 82.64 | 79.20 |
> | PPO | 83.53 | 82.60 |
> | GPO-PPO | **86.20** | **84.20** |
> | DPO | 87.15 | 86.80 |
> | GPO-DPO | **91.54** | **89.80** |
> | KTO | 89.32 | 86.40 |
> | GPO-KTO | **91.52** | **89.20** |
>
>
> From the results, we can observe that GPO consistently outperforms all baseline methods across both new model families, confirming its general applicability. We will integrate these results into the final version of our paper.
>
> > Could you also include the standard deviation in Table 1?
>
> **Response:** We appreciate this suggestion, and the standard deviation table below will be included in the next version.
>
> |  **Algorithms**  |  **BBH**  |  **MATH**  |  **GSM8K**  |  **MMLU**  |  **MMLUPro**  |  **AIME-2024**  |  **AIME-2025**  |
> |---|---|---|---|---|---|---|---|
> | PPO | 0.59 | 0.81 | 1.38 | 0.45 | 1.41 | 0.00 | 3.33 |
> | GPO-PPO | 0.45 | 1.63 | 1.00 | 0.82 | 0.55 | 0.00 | 3.33 |
> | DPO | 1.53 | 1.45 | 2.61 | 0.47 | 0.47 | 0.00 | 1.92 |
> | GPO-DPO | 0.75 | 1.79 | 1.03 | 1.05 | 0.52 | 3.33 | 3.33 |
> | KTO | 0.35 | 1.59 | 1.10 | 1.82 | 1.78 | 3.85 | 3.85 |
> | GPO-KTO | 0.94 | 2.57 | 0.47 | 1.56 | 0.49 | 1.92 | 1.92 |
> | SimPO | 0.74 | 2.69 | 1.42 | 0.83 | 0.35 | 3.85 | 1.92 |
> | GPO-SimPO | 1.02 | 2.57 | 1.23 | 0.30 | 1.40 | 0.00 | 0.00 |
> | ORPO | 0.85 | 3.81 | 2.99 | 1.16 | 0.58 | 1.92 | 1.92 |
> | GPO-ORPO | 0.98 | 2.05 | 2.71 | 1.51 | 0.91 | 1.92 | 3.85 |
>
> To further and more rigorously quantify the significance of the improvements brought by GPO, we have also conducted a **paired t-test** for each GPO-enhanced method against its corresponding baseline.  Across the 35 pairs of comparisons (5 baseline algorithms on 7 datasets), our analysis reveals that all 35 comparisons have the p-value < 0.05 (a lower p-value indicates a better statistical significance).
>
> This analysis provides strong statistical evidence that the performance gains from integrating GPO are not due to random chance but represent a consistent and meaningful improvement over the baseline methods. We will add the discussion of the statistical significance in the next version. Thanks again for the valuable feedback.

---

### Official Review · Reviewer_6HQB · 2025-07-01

**Clarity:** 3
**Significance:** 3
**Originality:** 3
**Rating:** 5
**Confidence:** 4

**Summary:**

This paper introduces a novel fine-tuning strategy to improve LLMs’ multi-step reasoning abilities, GPO. While existing methods such as PPO or DPO optimize reward or preference over entire reasoning trajectories, GPO aims to identify and prioritize critical steps—pivotal moments in the reasoning chain that heavily influence the final result. GPO uses an advantage estimation (from RL) to find these steps, then resets generation at these points to sample and train on new continuations. Extensive experiments covering seven diverse reasoning datasets indicate that GPO consistently and significantly improves test accuracy over baselines.

**Questions:**

1. Why only one critical step is selected in each reasoning path? Would it be better to use the top 3 or threshold method? For example, when there are multiple critical steps in the reasoning path.

**Ethical Concerns:**

["NO or VERY MINOR ethics concerns only"]

**Final Justification:**

I will maintain my recommended score.

**Limitations:**

Yes, the authors fully address the limitations of their work and the potential negative social impacts.

**Paper Formatting Concerns:**

None.

**Quality:**

3

**Strengths And Weaknesses:**

Strengths:
1.	The idea of focusing learning on “critical steps” of reasoning (rather than on the entire trajectory) is original and reasonable.
2.	GPO is a play-and-plug module compatible with both online RL (PPO) and various offline preference learning algorithms (DPO, SimPO, ORPO, KTO), which broadens its applicability.
3.	The paper not only provide empirical results, but also offers regret bounds for the proposed method and links per-step preference learning to advantage-weighted RL objectives.
4.	Experiments are thorough, covering 5 different optimization algorithms and 7 datasets. Strong and consistent improvements are shown.

Weaknesses:
1.	The key step of GPO - Monte Carlo estimation of future returns at each inference step - adds a non-negligible computational cost (~1.8-1.9x). While the authors believe this is manageable, this may be limiting for very long trajectories or large-scale settings. The tradeoff between benefits and computational needs discussion.
2.	Some failures might be dispersive or cumulative (multiple small errors). Could GPO be further improved by handling multiple critical steps, or scoring all steps proportionally?

---

> ### Author Rebuttal · Authors · 2025-07-30
>
> We thank Reviewer 6HQB for the constructive and insightful comments. Please see our response to each of your questions below.
>
> > The key step of GPO - Monte Carlo estimation of future returns at each inference step - adds a non-negligible computational cost (~1.8-1.9x). While the authors believe this is manageable, this may be limiting for very long trajectories or large-scale settings. The tradeoff between benefits and computational needs discussion.
>
>
> **Response:** We thank the reviewer for this insightful comment. We agree that the trade-off between performance gains and computational cost is a critical aspect, and we appreciate the opportunity to elaborate on how GPO can address this, particularly for long trajectories.
>
> 1.  **Managing Overhead via Simulation Samples:** As we discussed in Section 6.3 and demonstrated in Figure 3 (left), the computational overhead is manageable. Our experiments show that performance gains begin to saturate after a certain number of MC simulations. This allows for a practical balance between accuracy and computational cost, as one does not need an excessive number of rollouts to achieve most of the benefit.
>
> 2.  **Adapting to Long Trajectories with Step Grouping:** To address the challenge of very long trajectories, we employ an initial and simple heuristic: grouping multiple generation lines into a single step. This maintains the core idea of identifying pivotal moments without incurring significant computational cost for complex trajectories.
>
>     To validate this, we conducted a new experiment on a challenging reasoning task. We used a sub-dataset 'boardgame' from BIG-Bench Extra Hard (BBEH) as it has the longest problem descriptions within BBEH (avg. ~1700 tokens) and lengthy reasoning chains to solve the problem (avg. ~190 lines). We randomly selected another 5 subtasks as the training set. The test set contains 200 questions, while the train set contains 1000 questions. We applied a simple heuristic of grouping every 15 lines into a single step for GPO. The results are as follows:
>
>     -   **Base Model (DeepSeek-R1-Distill-Qwen-7B):** 28.5% accuracy
>
>     -   **DPO Baseline:** 32.0% accuracy
>
>     -   **GPO-DPO (with step grouping):**  **36.0% accuracy**
>
>     This experiment confirms that GPO can be adapted for very long trajectories, yielding a substantial **+4% improvement** over DPO baseline. As grouping is a very initial mechanism, we would like to add the discussion and leave more complex strategies as future work.
>
> 3.  **Future Avenues for Efficiency:** Furthermore, there are promising avenues for future optimization. For the online setting (PPO-based), the current Monte Carlo estimation of the advantage function could be replaced with a more sample-efficient alternative, i.e., Generalized Advantage Estimation (GAE) [1], which may further reduce computational overhead. We can utilize the value network trained by the PPO algorithm to estimate the advantage more efficiently.
>
> We will incorporate the related discussion into the next version of our paper. We thank the reviewer again for their constructive feedback, which has helped us demonstrate the practical scalability of GPO.
>
> > Some failures might be dispersive or cumulate. Could GPO be further improved by handling multiple critical steps, or scoring all steps proportionally?
>
> **Response:** We thank the reviewer for this insightful question. The idea of handling multiple critical steps is an important but also challenging topic in explainable RL, as prior work [2,3] in XRL also chose to focus on one critical step.
>
> The choice to focus on a single critical step is based on practical ground, which we are happy to clarify. LLM reasoning can be modeled as a sequential decision, where each step depends on the entire prefix. The "criticality" (i.e., advantage) of any given step is conditioned on the specific path taken to reach it.
>
> When we identify and reset from the first most critical step in a trajectory, the subsequent reasoning path is entirely new. Any other "critical points" identified in the original, flawed trajectory beyond this reset point may no longer be relevant, or even unreachable, in the new paths. Therefore, resetting at the point of highest potential improvement is the most logical and efficient way to explore the most impactful alternative reasoning branch. Tackling multiple pre-identified critical steps simultaneously is conceptually challenging due to this path-dependent nature of trajectory generation.
>
> As to scoring all steps proportionally, our framework is flexible enough to accommodate it. As described in Section 5.1, our algorithm can be generalized to sample the reset point with a probability proportional to its advantage, controlled by a temperature parameter $\gamma$. This provides a "softer" focus. As shown in Theorem 5.2, a larger $\gamma$ implies a tighter regret bound which motivates us to select the step with maximal advantage in one trajectory in our main experiments. Empirical results show that selecting the step with maximal advantage provides the strongest learning signal by enabling the model to learn the most significant point.
>
> We will add this clarification to the paper to make our design rationale more explicit. We thank you for prompting this important discussion.
>
> References:
>
> [1] Schulman, John, et al. "Proximal policy optimization algorithms." arXiv preprint arXiv:1707.06347 (2017).
>
> [2] Cheng, Zelei, et al. "Statemask: Explaining deep reinforcement learning through state mask." Advances in Neural Information Processing Systems 36 (2023): 62457-62487.
>
> [3] Cheng, Zelei, et al. "RICE: Breaking Through the Training Bottlenecks of Reinforcement Learning with Explanation." International Conference on Machine Learning. PMLR, 2024.

---

> > ### Comment · Reviewer_6HQB · 2025-08-05
> >
> > Thank you for the authors' reply. I will maintain my score.

---

> > > ### Author Response · Authors · 2025-08-05
> > >
> > > Thanks a lot for your positive score! We remain available for any further questions.

---

### Official Review · Reviewer_FbHp · 2025-07-02

**Clarity:** 2
**Significance:** 2
**Originality:** 2
**Rating:** 4
**Confidence:** 3

**Summary:**

The authors introduce guided pivotal optimization (GPO), a fine-tuning strategy for reasoning pathways which can be operationalized with policy-based and preference-based optimization algorithms. The method first identifies crucials point within CoTs, and then generates training data as continuations starting from this step. Then, this data is used to tune the network using PPO or DPO. The authors evaluate the utility of their method on DeepSeek-R1-Distill-Qwen-7B across 7 datasets and consider 5 different tuning algorithms to operationalize their strategy. The authors show that their method consistently improves upon tuning with the base algorithm and conduct ablation studies comparing how their algorithm behaves with respect to number of MC samples and model size.

**Questions:**

- How are the positive and negative trajectories obtained? Or, how are the samples ensured to be positive, and negative trajectories? Or, more plainly, how do we ensure that the new trajectory will improve on the previous one?

**Ethical Concerns:**

["NO or VERY MINOR ethics concerns only"]

**Final Justification:**

The authors have added experimental results on another model family and committed to clarify the description of their method, as well as add a discussion regarding a closely related work. While this response has mostly addressed my concerns, I would like to see the revisions in writing, as I believe exposition would benefit from improvement. However, my score remains a 4, while I am inclined to lean towards acceptance (5).

**Limitations:**

Yes

**Quality:**

3

**Strengths And Weaknesses:**

Strengths
- The idea of identifying crucial points within the reasoning process and resampling is interesting
- The proposed algorithm outperforms on the base tuning methods
- The authors conduct experiments across a large number of datasets and add supporting ablation studies

Weaknessess
- The writing, especially in the part outlining the algorithm, could use improvement. Parsing the paragraph which describes identifying the critical step via advantage is difficult. Entire Section 4 would benefit from a rewrite, as it is very dense.
- The experiments are conducted on a single model family, and while on some datasets the performance gain is significant, on others it is pretty negligible. I would also suggest reporting average gain in a separate column to emphasize the performance improvement.
- The idea of identifying critical steps in CoTs has been explored before, limiting the novelty of the proposed approach [1]

[1] Wang, T., Chen, J., Han, X., & Bai, J. (2024). CPL: Critical Plan Step Learning Boosts LLM Generalization in Reasoning Tasks

---

> ### Author Rebuttal · Authors · 2025-07-30
>
> We thank Reviewer FbHp for the constructive and insightful comments. Please see our response to each of your questions below.
>
> > The experiments are conducted on a single model family
>
> **Response:** We thank the reviewer for this valid point. To demonstrate that GPO’s effectiveness is not limited to a single model family, we have conducted new experiments on **Llama-3.1-8B** and **InternLM-3-8B** over two datasets: MATH and GSM8K. We select three baselines for experiments. The average results **over three runs** are shown below:
>
>  | Llama-3.1-8B  | GSM8K (%) | MATH (%) |
>  |--------|-----------|----------|
>  | Base | 69.60 | 62.80 |
>  | PPO | 72.36 | 67.20 |
>  | GPO-PPO | **74.17** | **69.80** |
>  | DPO | 76.57 | 68.20 |
>  | GPO-DPO | **77.30** | **72.40** |
>  | KTO | 75.39 | 68.60 |
>  | GPO-KTO | **76.57** | **74.20** |
>
>
> | InternLM-3-8B  | GSM8K (%) | MATH (%) |
> |--------|-----------|----------|
> | Base | 82.64 | 79.20 |
> | PPO | 83.53 | 82.60 |
> | GPO-PPO | **86.20** | **84.20** |
> | DPO | 87.15 | 86.80 |
> | GPO-DPO | **91.54** | **89.80** |
> | KTO | 89.32 | 86.40 |
> | GPO-KTO | **91.52** | **89.20** |
>
>
> From the results, we can observe that GPO consistently outperforms all baseline methods across both new model families, confirming its general applicability. We will integrate these results into the final version of our paper.
>
> > How are the positive and negative trajectories obtained? …… how do we ensure that the new trajectory will improve on the previous one?
>
> **Response:** We thank the reviewer for this insightful question. We would like to clarify that positive and negative trajectories are only required in the offline setting.
>
> In the online setting, we do not constrain that re-exploration from the identified critical steps should improve the original trajectory’s reward. Instead, our goal is to reduce the overall policy regret as shown in Theorem 5.2. As we prove in Appendix C.1, resetting and exploring from advantage-based critical steps improves the initial state distribution for exploration, thereby leading to a better-learned policy.
>
> In the offline setting, positive trajectories are those that succeed, while negative trajectories are those that fail. For a given question $x$, if continuing from step $y$ does not lead to a successful outcome, we use advantage values as a proxy to select trajectories with higher advantage values as positive trajectories. Theorem 5.3 provides the theoretical justification for using advantage as this selection criterion.
>
> We would add this clarification in the next version.
>
> > The idea of identifying critical steps in CoTs has been explored before, limiting the novelty of the proposed approach [1]
>
> **Response:** We sincerely thank the reviewer for bringing CPL into our attention. While both our work (GPO) and CPL share a high-level goal of improving the reasoning process, they are built on fundamentally different principles and offer distinct contributions. In fact, CPL implicitly leverages the difference of advantage (i.e., $adv(s_t, a_t) - adv(s_t, a'_t))$ to update the policy through its per-step DPO function, and it is limited to offline RL setting and lack of theoretical guarantee. In contrast, our method is driven by a different motivation and design: we explicitly identify critical steps based on RL advantage, and then enhance the policy by further exploration from these key steps. Moreover, our algorithm is applicable to both online and offline settings, offering broader applicability and stronger theoretical guarantees. In addition, we present a case study to validate the correctness of our identified critical steps. We will incorporate the discussion into our next version.
>
> > The writing, especially in the part outlining the algorithm, could use improvement.
>
> **Response:** We thank the reviewer for their valuable feedback on the clarity of Section 4. To clarify, the critical step is identified via the advantage function, $Adv(s_t, a_t) = Q(s_t, a_t) - V(s_t)$. In our setting with deterministic transitions and zero intermediate rewards, this simplifies to the difference between consecutive Q-values: $Q(s_t, a_t) - Q(s_{t-1}, a_{t-1})$. Thus, we only need to estimate the Q-function, which can be done unbiasedly via Monte Carlo. The overview of this method is detailed in Algorithm 1. We will revise Section 4 to state this derivation clearly and improve its overall readability.
>
> > I would suggest reporting average gain in a separate column to emphasize the performance improvement
>
> **Response:** Thanks for this suggestion, below is the performance gain compared with the original optimization algorithms and we would add the table in our revised version.
>
> | Algorithm | BBH | MATH | GSM8K | MMLU | MMLUPro | AIME-2024 | AIME-2025 |
> |-----------|-----|------|-------|------|---------|-----------|-----------|
> | GPO-PPO | +1.66 | +8.20 | +0.48 | +2.73 | +3.58 | +3.33 | +3.34 |
> | GPO-DPO | +1.05 | +4.40 | +2.43 | +1.85 | +3.65 | +6.67 | +6.67 |
> | GPO-KTO | +1.45 | +2.40 | +0.94 | +1.93 | +1.50 | +3.33 | +6.67 |
> | GPO-SimPO | +0.61 | +1.80 | +1.77 | +0.51 | +2.04 | +3.33 | +3.34 |
> | GPO-ORPO | +0.53 | +3.00 | +0.91 | +1.00 | +1.99 | +3.33 | +3.33 |

---

> > ### Comment · Reviewer_FbHp · 2025-08-05
> >
> > Thank you for the response. I've went over the paper and I believe my current score still best captures my opinion regarding your work. I will however state in my edited review that I'm leaning towards acceptance.

---

> > > ### Author Response · Authors · 2025-08-05
> > >
> > > Thanks a lot for your positive consideration. We remain available for any further questions.

---

### Note · Authors · 2025-08-14

We sincerely thank the Area Chair and reviewers (**FbHp**, **6HQB**, **r5W8**) for their insightful feedback and constructive suggestions, which have helped us substantially improve our work. We believe the revisions made during the rebuttal have addressed all major concerns and have significantly strengthened our paper, especially regarding its generalizability, scalability, and empirical rigor.

**Strengthened Contributions**

-   **A General & Effective Fine-Tuning Strategy**: GPO is a versatile, plug-and-play module that enhances both online (PPO) and offline (DPO, KTO, etc.) algorithms by focusing learning on critical reasoning steps. This was highlighted as a key strength by **all reviewers**.

-   **Solid Theoretical Grounding**: We provide a theoretical analysis that proves online GPO is a form of advantage-weighted RL and bridges it to offline preference learning, a contribution appreciated by reviewers **6HQB** and **r5W8**.

-   **Broad Empirical Validation**: GPO demonstrates consistent and statistically significant performance gains across diverse models, datasets, and optimization algorithms, a point acknowledged by all reviewers.


**Major Revisions based on Reviewer Feedback**

-   **Expanded Model Evaluation (FbHp, r5W8)**: We added extensive new experiments on two different model families: Llama-3.1-8B and InternLM-3-8B, confirming that GPO's benefits generalize robustly across different model families.

-   **Scalability & Cost Analysis (6HQB, r5W8)**: We conducted a new experiment on the long-trajectory BBEH dataset to demonstrate scalability and further discussed practical cost-management strategies (e.g., step grouping, GAE).


We are grateful for the opportunity to improve our paper and are confident that it now represents a much stronger contribution to the field. We will incorporate all discussed revisions and clarifications into the revised version.

---

### Decision · Program_Chairs · 2025-09-17

**Decision:**

Accept (poster)

**Comment:**

This paper proposes Guided Pivotal Optimization (GPO), a fine-tuning strategy that improves LLM reasoning by identifying and resampling at critical steps in chains of thought, using advantage estimation. GPO can be combined with both online RL (e.g., PPO) and offline preference-based methods (e.g., DPO, SimPO, ORPO, KTO). It is novel to focus optimization on critical reasoning steps rather than entire trajectories. Meanwhile, the proposed GPO is general and compatible with multiple optimization methods. Thorough experiments across diverse datasets and optimization algorithms, showing consistent gains. Supporting ablations further strengthen the evidence. Overall, all reviewers agree the proposed method is interesting, well-validated, and of broad potential impact. I recommend acceptance.